# Autoinhibition and regulation by phosphoinositides of ATP8B1, a human lipid flippase associated with intrahepatic cholestatic disorders

Thibaud Dieudonné[1,2], Sara Abad Herrera[3†], Michelle Juknaviciute Laursen[2†], Maylis Lejeune[1‡], Charlott Stock[2], Kahina Slimani[1], Christine Jaxel[1], Joseph A Lyons[4], Cédric Montigny[1], Thomas Günther Pomorski[3,5]*, Poul Nissen[2]*, Guillaume Lenoir[1]*

[1]Université Paris-Saclay, CEA, CNRS, Institute for Integrative Biology of the Cell (I2BC), Gif-sur-Yvette, France; [2]DANDRITE, Nordic EMBL Partnership for Molecular Medicine, Department of Molecular Biology and Genetics, Aarhus University, Aarhus, Denmark; [3]Faculty of Chemistry and Biochemistry, Department of Molecular Biochemistry, Ruhr University Bochum, Bochum, Germany; [4]Department of Molecular Biology and Genetics, Aarhus University, Aarhus, Denmark; [5]Department of Plant and Environmental Sciences, University of Copenhagen, Frederiksberg, Denmark

*For correspondence:
thomas.guenther-pomorski@ruhr-uni-bochum.de (TGüntherP);
pn@mbg.au.dk (PN);
guillaume.lenoir@i2bc.paris-saclay.fr (GL)

†These authors contributed equally to this work

Present address: ‡Institut Pasteur, Université de Paris, CNRS UMR3528, Structural Bioinformatics Unit, Paris, France

Competing interest: The authors declare that no competing interests exist.

**Abstract** P4-ATPases flip lipids from the exoplasmic to the cytosolic leaflet, thus maintaining lipid asymmetry in eukaryotic cell membranes. Mutations in several human P4-ATPase genes are associated with severe diseases, for example in *ATP8B1* causing progressive familial intrahepatic cholestasis, a rare inherited disorder progressing toward liver failure. ATP8B1 forms a binary complex with CDC50A and displays a broad specificity to glycerophospholipids, but regulatory mechanisms are unknown. Here, we report functional studies and the cryo-EM structure of the human lipid flippase ATP8B1-CDC50A at 3.1 Å resolution. We find that ATP8B1 is autoinhibited by its N- and C-terminal tails, which form extensive interactions with the catalytic sites and flexible domain interfaces. Consistently, ATP hydrolysis is unleashed by truncation of the C-terminus, but also requires phosphoinositides, most markedly phosphatidylinositol-3,4,5-phosphate (PI(3,4,5) $P_3$), and removal of both N- and C-termini results in full activation. Restored inhibition of ATP8B1 truncation constructs with a synthetic peptide mimicking the C-terminal segment further suggests molecular communication between N- and C-termini in the autoinhibition and demonstrates that the regulatory mechanism can be interfered with by exogenous compounds. A recurring (G/A)(Y/F)AFS motif of the C-terminal segment suggests that this mechanism is employed widely across P4-ATPase lipid flippases in plasma membrane and endomembranes.

## Editor's evaluation

This manuscript reports the first high-resolution structure of the P4 flippase ATP8B1, which is associated with intrahepatic cholestatic disorder in humans. Using biochemical studies guided by the structure, the authors demonstrate ATP8B1's autoinhibition mechanism, its regulation by lipids and phosphorylation, and a plausible mechanism of disease-associated mutation. These results are an important contribution to the expanding literature on membrane protein dynamics and function.

## Introduction

Transbilayer lipid asymmetry is a fundamental characteristic of eukaryotic cell and organelle membranes (*Kobayashi and Menon, 2018*; *van Meer, 2011*; *van Meer et al., 2008*; *Verkleij et al., 1973*). In most cell types choline-containing phosphatidylcholine (PC) and sphingomyelin (SM) are chiefly located in the exoplasmic leaflet while the aminophospholipids phosphatidylserine (PS) and phosphatidylethanolamine (PE), as well as phosphoinositides (PPIns), mostly occupy the cytoplasmic leaflet (*Bretscher, 1972*; *Murate et al., 2015*). Phospholipid asymmetry plays an important role in eukaryotic cell function. A well-studied example is the asymmetric distribution of PS in membranes of the late secretory/endocytic pathways, where it confers a high surface charge to these membranes, thereby facilitating the recruitment of polybasic motif-containing protein effectors such as the small G proteins K-Ras (*Hancock et al., 1990*; *Yeung et al., 2009*), Cdc42 and ROP6, as well as other proteins like protein kinase C (PKC), synaptotagmin, and the membrane fission protein EHD1 (*Bohdanowicz and Grinstein, 2013*; *Lee et al., 2015*; *Lemmon, 2008*; *Leventis and Grinstein, 2010*; *Platre et al., 2019*). Thus, there is a direct link between PS sidedness and regulation of cell polarity, cell signaling and vesicular trafficking. Phospholipid asymmetry is maintained by flippases and floppases, which use ATP for inward and outward movement of lipids across membranes, respectively (*Andersen et al., 2016*; *López-Marqués et al., 2015*; *Montigny et al., 2016*). In contrast, scramblases comprise a third category that passively equilibrates lipids across the bilayer, often controlled by gating (*Pomorski and Menon, 2016*). Although floppases belong to the superfamily of ATP-binding cassette (ABC) transporters, most flippases characterized thus far are from the type 4 subfamily of P-type ATPases, hereafter referred to as P4-ATPases. The human genome encodes 14 P4-ATPases. Using NBD-lipids as fluorescent derivatives of native lipids, ATP8A1, ATP8A2, ATP11A, ATP11B and ATP11C were shown to transport the aminophospholipids NBD-PS and NBD-PE, both in cell-based assays and upon reconstitution in proteoliposomes (*Coleman et al., 2009*; *Lee et al., 2015*; *Segawa et al., 2016*; *Wang et al., 2018*). By contrast, ATP8B1, ATP8B2, and ATP10A were shown to transport NBD-PC (*Naito et al., 2015*; *Takatsu et al., 2014*) and ATP10A and ATP10D catalyze the transport of NBD-glucosylceramide (*Roland et al., 2019*). Mutations in ATP8A2 and ATP11A have been reported to cause severe neurological disorders (*Onat et al., 2013*; *Segawa et al., 2021*), and mutations in ATP8B1 are associated with intrahepatic cholestatic disorders, such as benign recurrent intrahepatic cholestasis (BRIC1), intrahepatic cholestasis of pregnancy (ICP1), and the more severe progressive familial intrahepatic cholestasis type 1 (PFIC1). PFIC1 is a rare inherited liver disorder characterized by impaired bile flow, fat malabsorption and progressive cirrhosis and fibrosis (*Jacquemin, 2012*; *van der Mark et al., 2013*).

Similar to many ion-transporting P-type ATPases, P4-ATPases consist of a transmembrane domain containing ten membrane-spanning α-helical segments, as well as three cytosolic domains, the actuator (A), nucleotide-binding (N), and phosphorylation (P) domains involved in catalysis (*Figure 1A*). Importantly, most P4-ATPases form obligatory binary complexes with members of the CDC50 protein family, which are essential for correct targeting of the flippase complex to its final destination and for its transport activity (*Coleman and Molday, 2011*; *Lenoir et al., 2009*; *Poulsen et al., 2008*; *Saito et al., 2004*; *Segawa et al., 2018*). Conformational changes in the membrane domain, required to facilitate lipid transport, are coupled to phosphorylation and dephosphorylation events in the cytosolic ATPase domains, thereby allowing efficient lipid transport against concentration gradients. The different steps of the transport cycle are collectively described as the Post-Albers scheme (*Albers, 1967*; *Post et al., 1972*), where the P-type ATPase cycles between different conformations, E1, E1P, E2P, and E2 (P for phosphorylated) (*Figure 1B*). The transport substrate, a lipid for P4-ATPases, is recognized in the E2P conformation, and its binding triggers dephosphorylation leading to E2 and eventually release of the lipid in the opposing leaflet. The subcellular localization, heteromeric interactions with CDC50 proteins and lipid transport activity of ATP8B1 have been thoroughly investigated using cell-based assays (*Bryde et al., 2010*; *Takatsu et al., 2014*; *van der Velden et al., 2010*). In contrast, ATP8B1 remains poorly studied at the molecular mechanistic level. In particular, while several other P4-ATPases, and P-type ATPases in general, are tightly regulated by lipid co-factors, protein partners, or by their terminal extensions (*Azouaoui et al., 2017*; *Chalat et al., 2017*; *Holemans et al., 2015*; *Saffioti et al., 2021*; *Tsai et al., 2013*), the way ATP8B1 activity is regulated remains unknown. Recent high-resolution structures of the yeast Drs2-Cdc50, Dnf1,2-Lem3 and the human ATP8A1-CDC50A and ATP11C-CDC50A flippase complexes have illuminated the molecular mechanism of lipid

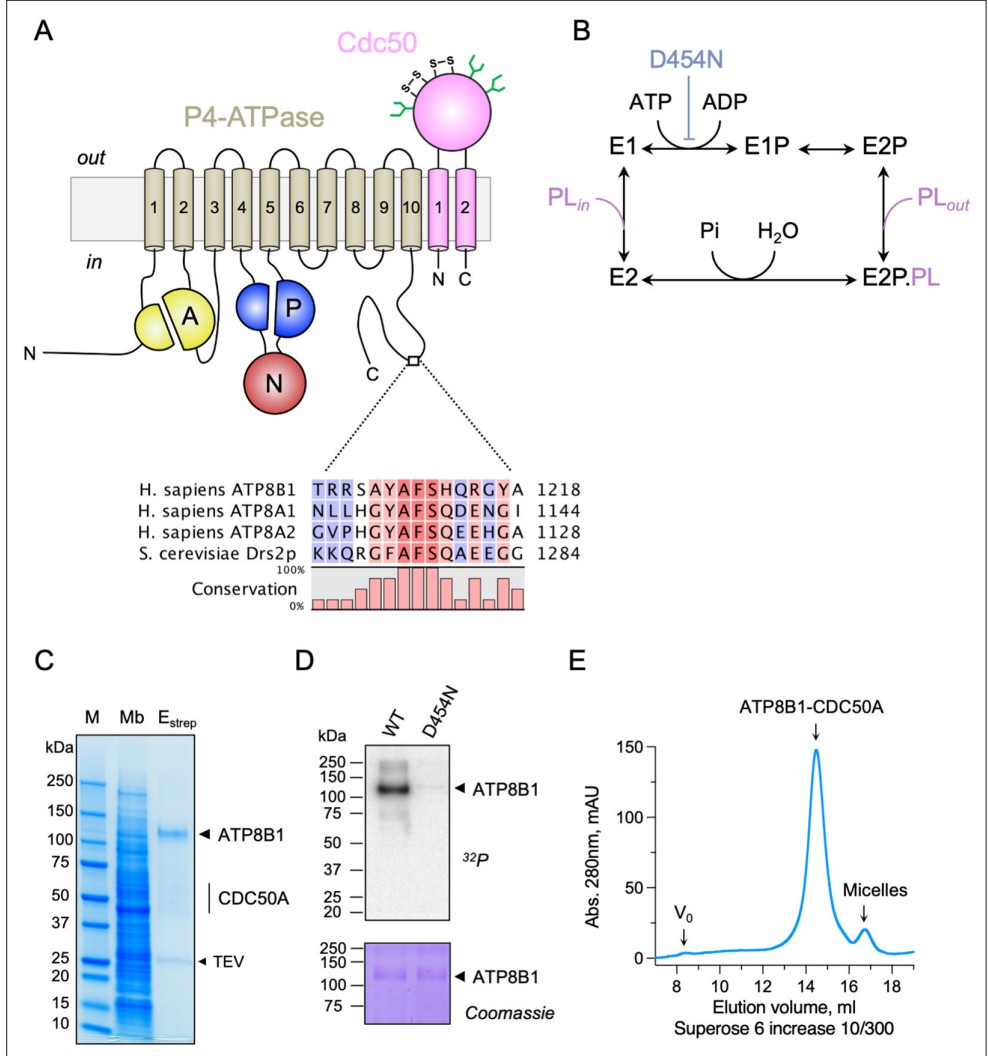

**Figure 1.** Purification and functional assessment of the ATP8B1-CDC50A complex expressed in *Saccharomyces cerevisiae*. (**A**) Predicted topology of ATP8B1-CDC50A with the transmembrane domain of ATP8B1 in tan and the Actuator domain (**A**), the Nucleotide binding domain (**N**) and the Phosphorylation domain (**P**) in yellow, red, and blue, respectively. CDC50A with two transmembrane spans and a large exoplasmic loop is in pink; predicted disulphide bridges (**S–S**) and glycosylation sites (green) are indicated. Sequence alignment of part of the C-terminus of ATP8B1, ATP8A1, ATP8A2, and Drs2 (CLC Main Workbench, Qiagen). The shading indicates conservation (blue 0% – red 100%). Uniprot accession numbers are P39524 for Drs2, Q9Y2Q0 for ATP8A1, Q9NTI2 for ATP8A2 and O43520 for ATP8B1. (**B**) Post-Albers cycle for P4-ATPases. ATP8B1 mutation D454N prevents phosphorylation on the catalytic aspartate and thus blocks activity. Pi, inorganic phosphate; PL, phospholipid. (**C**) SDS-PAGE analysis of ATP8B1-CDC50A affinity purification on streptavidin beads. Crude yeast membranes (Mb), containing 25 µg of total proteins, of which ATP8B1 represents 0.5%, and ~1–1.5 µg proteins recovered upon TEV protease cleavage on streptavidin beads (E$_{strep}$) were loaded on the gel and visualized by Coomassie Blue staining. M, molecular weight marker. (**D**) Phosphoenzyme formation from [γ-$^{32}$P]ATP of wild-type and catalytically-inactive D454N variant, as analyzed after electrophoretic separation on acidic gels. Coomassie Blue staining of the same gel was used to control the amount of wild-type and D454N subjected to $^{32}$P labeling. (**E**) Size-exclusion chromatography elution profile of the purified human ATP8B1-CDC50A complex used for cryo-EM studies. Arrows indicate the void volume of the column (**V$_0$**), as well as the elution volume of the ATP8B1-CDC50A complex and detergent micelles.

The online version of this article includes the following source data and figure supplement(s) for figure 1:

**Figure supplement 1.** Strategy for purification of the ATP8B1-CDC50A complex.

**Figure supplement 1—source data 1.** GraphPad Prism tables for results displayed in *Figure 1—figure supplement 1E*.

transport, providing a framework for understanding how these transporters are able to move lipids (*Bai et al., 2019*; *Bai et al., 2020*; *Hiraizumi et al., 2019*; *Lyons et al., 2020*; *Nakanishi et al., 2020b*; *Timcenko et al., 2019*; *Timcenko et al., 2021*). A key finding from these high-resolution structures is C-terminal autoinhibition of yeast Drs2 and human ATP8A1 (*Hiraizumi et al., 2019*; *Timcenko et al., 2019*). Furthermore, structures of Drs2-Cdc50 obtained in the presence of phosphatidylinositol-4-phosphate (PI(4)P) shed light on the specific regulation of Drs2 by this phosphoinositide, as previously observed using purified enzyme and activity assays (*Azouaoui et al., 2017*; *Natarajan et al., 2009*; *Zhou et al., 2013*).

In this report, we purified human ATP8B1-CDC50A complex, amenable for detailed study of its three-dimensional structure and catalytic activity. We determined the structure at 3.1 Å resolution of an autoinhibited state by cryo-electron microscopy (cryo-EM). In keeping with an observed, tight interaction of the C-terminal tail of ATP8B1 with the cytosolic domains, the ATP8B1-CDC50A complex displayed ATPase activity only after removal of its C-terminus. Using protease cleavage sites within the N-terminus or, for the C-terminus, immediately after the last transmembrane segment of ATP8B1, we demonstrate that ATP8B1 is primarily autoinhibited by its C-terminal extension, but that the N-terminal extension is involved in a synergistic manner. In addition to the importance of these autoregulatory elements, we show that PPIns are critical activators of ATP8B1 activity.

## Results

### Cryo-EM structure of the ATP8B1-CDC50A complex in the autoinhibited E2P state

Recent studies revealed that flippases can be autoregulated by their C-terminal extensions. In particular, (G/A)(Y/F)AFS motifs in the C-termini of Drs2 and ATP8A1 occupy the nucleotide-binding site, thereby preventing conformational changes required for lipid transport (*Hiraizumi et al., 2019*; *Timcenko et al., 2019*). This motif is also present in ATP8B1 as $^{1208}$AYAF$^{1212}$S (*Figure 1A*), hinting at a regulatory role of the ATP8B1 C-terminus. To gain insight into the mechanism of ATP8B1 regulation, we devised a procedure for co-overexpression of ATP8B1 and CDC50A in *Saccharomyces cerevisiae* and purification of the complex (*Figure 1—figure supplement 1A B*) based on experiences from expression of the yeast Drs2 and Cdc50. ATP8B1 and CDC50A co-expressed well in yeast and were solubilized from yeast membranes using n-Dodecyl-β-ᴅ-Maltoside (DDM) supplemented with cholesteryl hemisuccinate (CHS). Following streptavidin-based affinity chromatography and on-column cleavage of the biotin acceptor domain (BAD) tag with TEV protease, we obtained a highly pure ATP8B1-CDC50A complex (*Figure 1C*, *Figure 1—figure supplement 1C*). Treatment of the purified ATP8B1-CDC50A complex with Endoglycosidase H resulted in consolidation of multiple bands into a single band around 40 kDa, the expected molecular weight of histidine-tagged CDC50A, reflecting various glycosylation levels of its polypeptide chain (*Figure 1—figure supplement 1C*). The stoichiometry between ATP8B1 and CDC50A was found to be 1:1, as determined by in-gel fluorescence (*Figure 1—figure supplement 1D E*). P-type ATPases couple autophosphorylation from ATP and subsequent dephosphorylation of a catalytic aspartate in the P-domain to structural changes in the membrane domain, thus transporting substrates across the membrane against steep concentration gradients (*Figure 1B*). To ascertain functionality of the purified complex, we investigated its ability to undergo phosphorylation from [γ-$^{32}$P]ATP on its catalytic aspartate. The results confirm that the phosphoenzyme involves formation of an aspartyl-phosphate bond on residue D454 (*Figure 1D*). For structural studies, DDM was exchanged for lauryl maltose neopentyl glycol (LMNG). The resulting sample showed a high degree of monodispersity on size-exclusion chromatography (*Figure 1E*).

The structure of the full-length complex was then determined using single particle cryo-EM (*Figure 2—figure supplement 1*). To stabilize the complex in the autoinhibited E2P conformation (E2P$_{autoinhibited}$), the sample was incubated in the presence of beryllium fluoride (forming e.g. BeF$_3^-$, BeF$_2$(OH$_2$) adducts, referred to as BeF$_x$) mimicking phosphorylation. The high-resolution map (overall resolution: 3.1 Å) obtained by cryo-EM enabled us to model most of ATP8B1 and CDC50A sequences (*Table 1*), except flexible loops and termini.

As expected, ATP8B1 harbors a typical P4-ATPase fold with a transmembrane helical bundle made of 10 α-helical segments, a nucleotide binding domain (N), a phosphorylation domain (P) and an actuator domain (A) (*Figure 2*). Comparison with other P4-ATPase structures and the presence of an

**Table 1.** Cryo-EM data collection, refinement, and validation statistics.

| Data collection and processing | |
| --- | --- |
| Magnification | ×130,000 |
| Voltage (kV) | 300 |
| Microscope | Titan Krios (Aarhus University) |
| Camera | Gatan K3 |
| Physical pixel size (Å/pix) | 0.66 |
| Electron exposure (e–/Å$^2$) | 60 |
| Defocus range (µm) | 0.7–1.8 |
| Number of movies | 3,918 |
| Initial particle images (no.) | 470,103 |
| Final particle images (no.) | 104,643 |
| Symmetry imposed | C1 |
| Map resolution (Å) | 3.1 |
| FSC threshold | 0.143 |
| Map resolution range (Å) | 2.7–4.5 |
| **Refinement** | |
| Initial model used (PDB code) | ATP8B1: I-TASSER homology model based on 6ROH<br>CDC50A: 6K7L |
| Model resolution (Å) | 3.3 |
| FSC threshold | 0.5 |
| Map sharpening $B$ factor (Å$^2$) | –84 |
| Model composition | |
| Non-hydrogen atoms | 11,868 |
| Protein residues | 1,439 |
| Ligands | 1 MG, 1 BEF, 4 Y01, 4 NAG, 1 BMA |
| $B$ factors (Å$^2$, min/max/mean) | |
| Protein | 33.89/136.87/67.09 |
| Ligand | 41.58/110.52/60.89 |
| R.m.s. deviations | |
| Bond lengths (Å) | 0.002 |
| Bond angles (°) | 0.492 |
| Validation | |
| MolProbity score | 1.43 |
| Clashscore | 4.74 |
| Poor rotamers (%) | 0.08 |
| Ramachandran plot | |
| Favored (%) | 96.92 |
| Allowed (%) | 3.08 |
| Disallowed (%) | 0.0 |

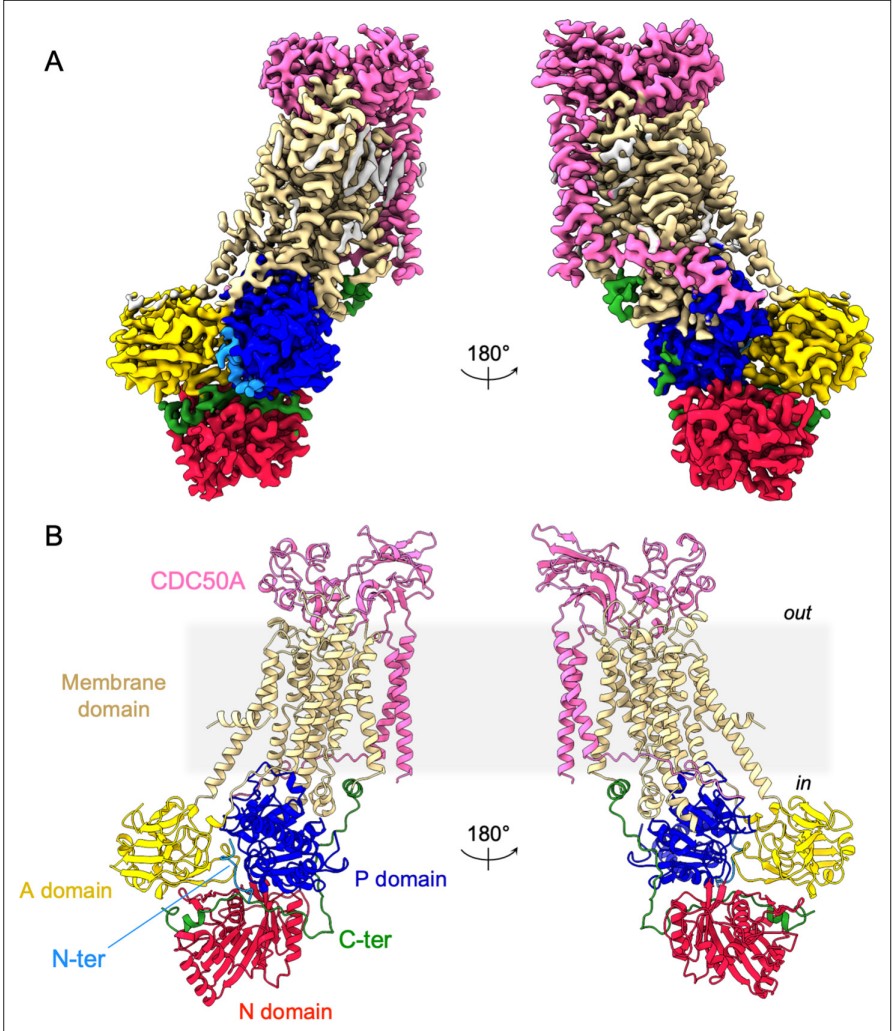

**Figure 2.** Overall ATP8B1-CDC50A structure. (**A**) Cryo-EM map of ATP8B1-CDC50A in the E2P autoinhibited state. The cytosolic A-, N-, and P-domains of ATP8B1 are colored in yellow, red and blue, respectively. The transmembrane domain of ATP8B1 is colored in tan. The N- and C-terminal tails of ATP8B1 are colored in cyan and green, respectively. CDC50A is colored in pink. CHS densities and residual densities corresponding to detergent or less ordered unmodelled lipids are in grey. (**B**) Cartoon representation of the refined model. Colors are as in (**A**). Electron microscopy data bank (EMDB) accession number: EMD-13711. Protein Data Bank (PDB) accession number: 7PY4.

The online version of this article includes the following figure supplement(s) for figure 2:

**Figure supplement 1.** Data processing flow chart.

**Figure supplement 2.** Cryo-EM density of the ATP8B1-CDC50A complex and its corresponding model.

**Figure supplement 3.** Structural comparison of P4-ATPases-CDC50A complexes of known structure.

extra density in the phosphorylation site confirmed that our structure resembles an E2P$_{autoinhibited}$ state with bound BeF$_x$ (*Figure 2—figure supplement 2*). Both CDC50A and CDC50B have been found to interact with ATP8B1 and to promote its trafficking to the plasma membrane (*Bryde et al., 2010*). As observed for other P4-ATPase-Cdc50 complexes (*Bai et al., 2019*; *Hiraizumi et al., 2019*; *Nakanishi et al., 2020b*; *Timcenko et al., 2019*), CDC50A interacts extensively with ATP8B1, through its extracellular, transmembrane, and N-terminal region. The extracellular domain of CDC50A covers all the extracellular loops of ATP8B1 except the TM1-2 loop while the N-terminal tail extends parallel to the membrane, interacting with TM6-7 and TM8-9 loops of ATP8B1, as well as with the segment linking TM4 to the P-domain, as previously described (*Hiraizumi et al., 2019*; *Timcenko et al., 2019*). The transmembrane domain of CDC50A is made of two interacting transmembrane α-helices and

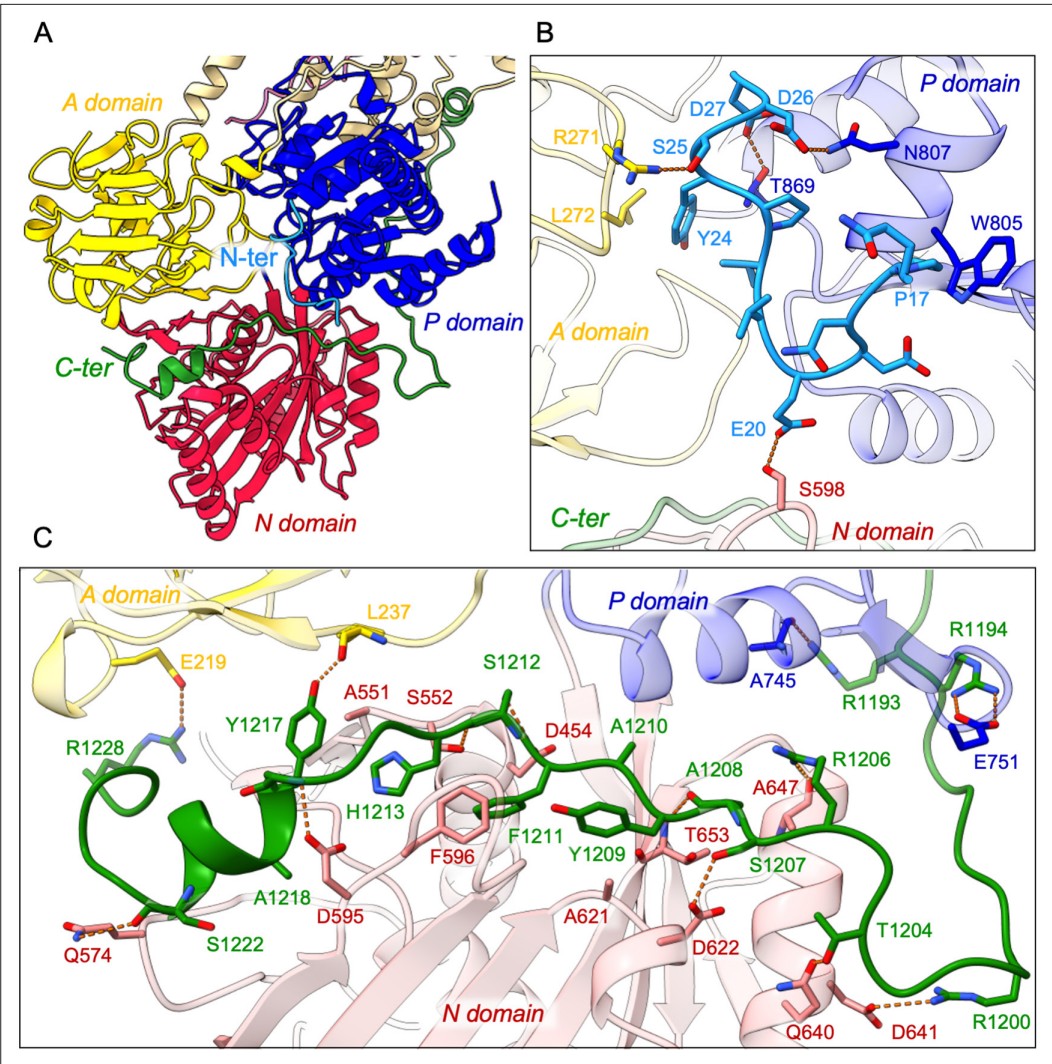

**Figure 3.** Detailed interaction of the N- and C-terminal tails with the cytosolic A-, N-, and P-domains of ATP8B1. (**A**) Overall view of the cytosolic A-, N-, and P-domains colored in yellow, red and blue, respectively. The transmembrane domain is colored tan. The N- and C-terminal tails of ATP8B1 are colored in cyan and green, respectively. (**B, C**) Close-up view highlighting the interactions between residues in the N-terminal tail and the cytosolic domains of ATP8B1 (**B**) or the C-terminal tail and the cytosolic domains of ATP8B1 (**C**). Electrostatic interactions are shown as orange dashes.

The online version of this article includes the following figure supplement(s) for figure 3:

**Figure supplement 1.** overall and close-up views of the N- and C-terminal extensions of ATP8B1 and their corresponding EM densities.

three N-linked glycosylation sites are clearly visible in the cryo-EM map (N107, N180, and N294), indicating that *S. cerevisiae* supports glycosylation of this human transporter. CDC50A exhibited a structure nearly identical to that observed in the ATP8A1-CDC50A and ATP11C-CDC50A human complexes (*Hiraizumi et al., 2019*; *Nakanishi et al., 2020a*), with a RSMD of 0.8 and 1.1 Å, respectively (*Figure 2—figure supplement 3*).

In addition, the cryo-EM data displayed very clear densities for parts of the N- and C-termini of ATP8B1 (*Figure 3A*, *Figure 3—figure supplement 1*). Interestingly, the N-terminal region (Q16-D27) was found to interact tightly with the P-, A- and the N-domain of ATP8B1 (*Figure 3B*). Regarding the interaction of the N-terminal tail with the P-domain, residues D26 and D27 are involved in electrostatic interactions with side chains of N807 and T869, respectively. The interaction is further enhanced by hydrophobic interactions between P17 and W805. The N-terminal tail interacts with the A-domain

through hydrogen bonds between S25 and R271 and is further reinforced by hydrophobic interaction between Y24 and L272. Finally, the interaction of the N-terminal tail with the N-domain is mediated by a hydrogen bond between E20 and S598 (*Figure 3B*). Similarly, the C-terminal tail of ATP8B1 engages in hydrogen bonds as well as several salt bridges and hydrophobic interactions with the three cytosolic domains (*Figure 3C*). Noteworthily, F1211 in the conserved AYAFS motif interacts via π-π interactions with F596 in the N-domain, which normally interacts with the adenosine ring of ATP in P-type ATPases, thereby preventing ATP binding. Hydrogen bonds between T1204-Q640, R1206-A647, S1207-D622, and S1212-S552 pairs further promote tight interaction between the C-terminal tail and the N-domain. Interactions of the C-terminal tail with the A- and P-domains are mediated by salt bridges (between R1228 and E219 and between R1194 and E751) or via hydrogen bonding between the side chains of Y1217 and R1193 with the backbone carbonyl groups of L237 and A745, respectively (*Figure 3C*).

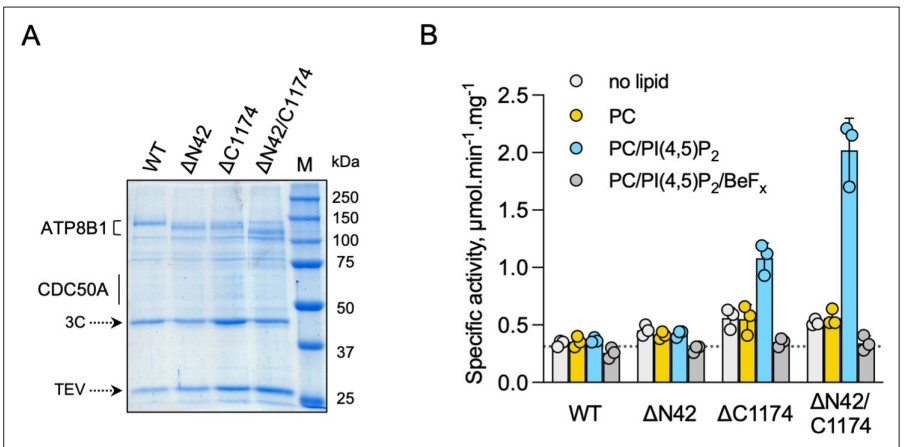

**Figure 4.** ATP8B1-CDC50A is autoinhibited by both its N- and C-terminal tails and the presence of lipids is required for its activity. (**A**) Removal of N- and/or C-terminal extensions of ATP8B1 upon on-column cleavage of streptavidin-bound ATP8B1-CDC50A with both TEV and 3 C proteases assessed by Coomassie blue stained SDS-PAGE. ΔN42 lacks residues 1–42 of ATP8B1 whereas ΔC1174 lacks residues 1175–1251 and ΔN42/C1174 lacks both. M, molecular weight marker. Streptavidin-purified wild-type (WT) and truncated mutants were used for subsequent ATPase assays. (**B**) ATPase activity of wild-type (WT), N-terminally truncated (ΔN42), C-terminally truncated (ΔC1174) and both N- and C-terminally truncated (ΔN42/C1174) ATP8B1 ( ~ 5 μg ml⁻¹ protein) in complex with CDC50A determined in DDM at 30 °C. The assay medium contained 1 mM MgATP, 0.5 mg ml⁻¹ DDM, and 0.01 mg ml⁻¹ CHS. PC and PI(4,5)P$_2$ were added at 0.1 mg ml⁻¹ (132 μM) and 0.025 mg ml⁻¹ (23 μM), respectively, resulting in a DDM final concentration of 1.25 mg ml⁻¹. The PC/PI(4,5)P$_2$ ratio is therefore 5.8 (mol/mol) Data are a mean ± s.d. of three technical replicate experiments (purification #1, see Materials and methods). The dotted line represents background NADH oxidation due to photobleaching, measured in the absence of purified protein and lipids. Source files related to (**B**) are available in *Figure 4—source data 1*.

The online version of this article includes the following source data and figure supplement(s) for figure 4:

**Source data 1.** GraphPad Prism tables for results displayed in *Figure 4B*.

**Figure supplement 1.** Sequence alignment of selected P4-ATPases.

**Figure supplement 2.** The amount of CDC50A which co-elutes with ATP8B1 upon on-column cleavage with TEV and 3 C proteases is similar for wild-type ATP8B1 (WT) and the 3 C protease cleavage site insertion mutants (ΔN42, ΔC1174, ΔN42/C1174).

**Figure supplement 3.** ATPase activity measurements of streptavidin-purified WT and catalytically-inactive D454N ATP8B1-CDC50A.

**Figure supplement 3—source data 1.** GraphPad Prism tables for results displayed in *Figure 4—figure supplement 3B*.

**Figure supplement 4.** Catalytic properties of the purified ATP8B1-CDC50A complex.

**Figure supplement 4—source data 1.** GraphPad Prism tables and curve fitting for results displayed in *Figure 4—figure supplement 4A, B*.

## Autoinhibition of ATP8B1 by its N- and C-termini

To investigate the role of ATP8B1 N- and C-termini, we inserted 3 C protease cleavage sites after residue P42 in the N-terminus, to remove most of the N-terminal tail including the Q16-D27 region found in the structure (ΔN42), and/or after residue E1174 at the end of the last transmembrane segment 10, to remove the C-terminus (ΔC1174 and ΔN42/C1174) (*Figure 4—figure supplement 1*). The various 3 C protease constructs were purified by streptavidin affinity chromatography (*Figure 4A*), with yields ranging from half (for ΔN42), to one fourth (for ΔC1174 and ΔN42/C1174) of that obtained for the wild-type (WT) complex. Noteworthy, insertion of the 3 C protease cleavage sites did not alter the interaction between ATP8B1 and CDC50A, as shown by immunoblotting of the fraction collected upon incubation of streptavidin beads with 3 C and TEV (*Figure 4—figure supplement 2*). Removal of the N-terminus and/or the C-terminus was not sufficient to stimulate ATP8B1-CDC50A ATPase activity in the presence of its transport substrate PC, suggesting an additional regulatory mechanism (*Figure 4B*). PI(4)P has been shown to be essential to stimulate ATP hydrolysis by Drs2, a yeast homolog of ATP8B1 (*Azouaoui et al., 2017*). Considering that ATP8B1 is localized at the plasma membrane (PM), we reasoned that addition of PI(4,5)$P_2$, the most abundant phosphoinositide in the PM (*Balla, 2013*; *Dickson and Hille, 2019*), might be required to elicit ATP8B1 activity. While PI(4,5)$P_2$ proved unable to stimulate the intact WT ATP8B1-CDC50A complex, limited proteolysis of the complex with trypsin dramatically increased the rate of ATP hydrolysis, consistent with autoinhibition of the intact ATP8B1-CDC50A complex (*Figure 4B*, *Figure 4—figure supplement 3*). We observed a ~ fourfold increase of the BeF$_x$-sensitive ATP hydrolysis upon addition of PI(4,5)$P_2$ for the C-terminally truncated construct (*Figure 4B*). Interestingly, removal of both termini resulted in additional activation of ATP8B1 suggesting that, although the sole removal of the N-terminus has seemingly no effect on autoinhibition relief, the N-terminus cooperates with the C-terminus for full autoinhibition of the ATP8B1-CDC50A complex (*Figure 4B*). Addition of BeF$_x$ inhibited the ATPase activity of ΔN42/C1174 ATP8B1 with an IC$_{50}$ of ~45 µM, consistent with the ability of this structural analog of phosphate to act as a general P-type ATPase inhibitor (*Figure 4—figure supplement 4A*; *Danko et al., 2009*). Finally, the purified ATP8B1-CDC50A complex showed a $K_m$ of ~40 µM for MgATP (*Figure 4—figure supplement 4B*).

We then asked whether addition of a peptide mimicking the C-terminus of ATP8B1 inhibited the activated enzyme. Of specific relevance, large-scale phosphoproteomic studies have shown that mouse ATP8B1 is phosphorylated at residue S1223 (*Huttlin et al., 2010*; *Villén et al., 2007*). Given that S1223 is conserved between mouse and human ATP8B1 and that this residue is located at the interface of the A- and the N-domain (*Figure 5A*), we used the non-phosphorylated and phosphorylated versions of the C-terminal peptide to more precisely assess the involvement of the ATP8B1 C-terminal region in autoinhibition and to address the effect of this putative phosphorylation on the autoinhibition mechanism. A peptide encompassing the AYAFS motif (residues 1205–1251, *Figure 4—figure supplement 1*) was chemically synthesized and incubated with ΔN42/C1174 ATP8B1. The C-terminal peptide efficiently inhibited ATP hydrolysis by ATP8B1, with an IC$_{50}$ of ~22 µM (*Figure 5B and D*, and *Table 2*), without adversely impacting proper functioning of the enzyme-coupled assay (*Figure 5—figure supplement 1*).

Remarkably, phosphorylation at S1223 impaired the ability of the C-terminal peptide to inhibit ΔN42/C1174 ATP8B1, with an IC$_{50}$ shifted to approximately 380 µM (*Figure 5B and D* and *Table 2*). Furthermore, inhibition of ΔC1174 ATP8B1, that is still containing the N-terminal tail, was about 270-fold more efficient (IC$_{50}$ ~0.08 µM) than ΔN42/C1174. Similar to the effect on the ΔN42/C1174 variant, phosphorylation at S1223 decreased the ability of the C-terminal peptide to inhibit ATPase activity of the ΔC1174 ATP8B1 variant (*Figure 5C and D* and *Table 2*). These results strongly support a prominent role for the N-terminal tail of ATP8B1 in the autoinhibition mechanism. Importantly, inhibition was specific as neither the yeast P4-ATPase Drs2, nor the cation-transporting Na$^+$/K$^+$-ATPase (a P2-ATPase), could be inhibited by the C-terminal tail of ATP8B1 (*Figure 5E*).

Together, our data reveal that the ATP8B1-CDC50A flippase is autoinhibited by its N- and C-terminal extensions in a cooperative mechanism and that PI(4,5)$P_2$ is a major regulator of its activity.

## Lipid-dependence of ATP8B1 activity

We showed that ATP8B1-CDC50A required PC and PI(4,5)$P_2$ for enzyme turnover (*Figure 4B*). We next explored the effect of other lipid species on the enzyme turnover in the presence of PI(4,5)$P_2$.

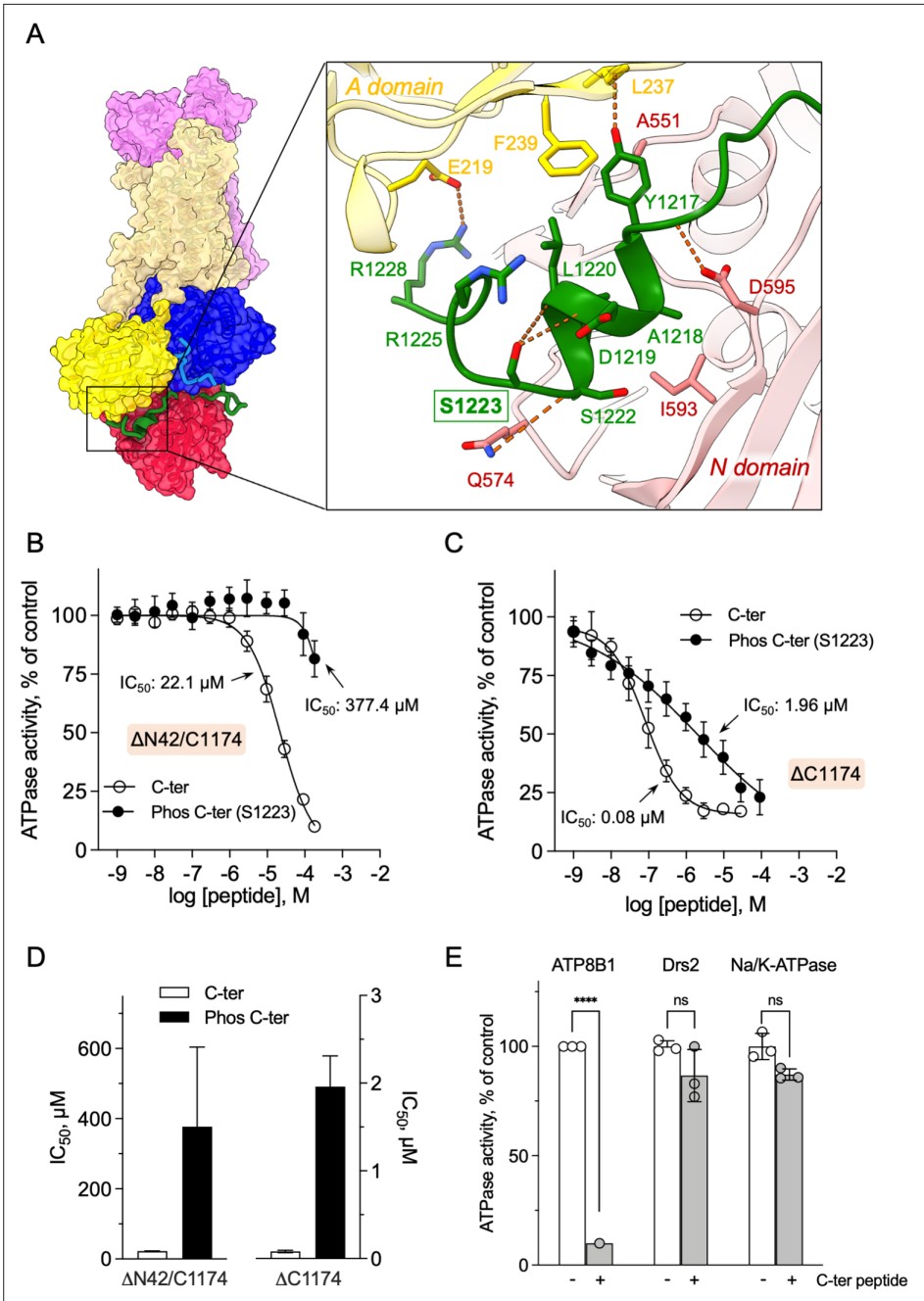

**Figure 5.** Autoinhibition of ATP8B1 by its N- and C-terminal extensions. (**A**) Overall and close-up views of S1223 in the cleft formed by the A- and N-domains. The cytosolic A- and N-domains of ATP8B1 are colored in yellow and red, respectively, and are shown as surface and cartoon. The C-terminal tail of ATP8B1 is shown as cartoon with side chains in green. Electrostatic interactions are shown as orange dashes. (**B**) Back-inhibition of ΔN42/ C1174 ATP8B1 ( ~ 3–3.3 µg ml⁻¹ protein) by synthetic C-terminal peptides (C-ter, Phos C-ter). ATPase activity was determined at 37 °C. The BeF$_x$-sensitive ATPase activity is plotted, taking the activity in the absence of the C-terminal peptide as 100%. The data were fitted to an inhibitory dose-response equation with variable slope. 95% confidence intervals for IC$_{50}$ values are given as CI[lower CI, upper CI]. C-terminal peptide: CI[$1.98 \times 10^{-5}$, $2.48 \times 10^{-5}$]; phosphorylated C-terminal peptide: CI[$1.44 \times 10^{-4}$, $9.90 \times 10^{-4}$]. Data are mean ± s.d. of three replicate experiments (purification #2, see Materials and methods). (**C**) Back-inhibition of ΔC1174 ( ~ 3–3.3 µg ml⁻¹ protein) by C-terminal peptides. ATPase activity was determined at 37 °C. The BeF$_x$-sensitive ATPase activity is plotted, taking the activity in the absence of C-terminal peptide as 100%. The data were fitted to an inhibitory dose-response equation with variable slope. C-terminal peptide: CI[$5.86 \times 10^{-8}$, $1.12 \times 10^{-7}$]; phosphorylated C-terminal

*Figure 5 continued on next page*

*Figure 5 continued*

peptide: CI[$1.40 \times 10^{-6}$, $2.73 \times 10^{-6}$].Data are a mean ± s.d. of three to four replicate experiments (purification #2, see Materials and methods). For panels (**B**) and (**C**), the assay medium contained 1 mM MgATP, 0.5 mg ml$^{-1}$ DDM, and 0.01 mg ml$^{-1}$ CHS. PC and PI(4,5)P$_2$ were added at 43 µg ml$^{-1}$ (43 µM) and 0.025 mg ml$^{-1}$ (23 µM), respectively. (**D**) Half-maximal inhibitory concentration (IC$_{50}$) of ATP8B1-CDC50A ATPase activity by C-terminal peptides deduced from curves in (**B**) and (**C**). Error bars represent the mean ± s.d. based on 33–47 data points. (**E**) Specificity of ATP8B1 inhibition by its C-terminal tail. ATPase activity of purified DDM-solubilized Drs2-Cdc50 (20 µg ml$^{-1}$) and pig α1β1 Na$^+$/K$^+$-ATPase (10 µg ml$^{-1}$) in microsomal membranes was determined at 30°C and 37°C, respectively, in the absence or presence of 180 µM ATP8B1 C-terminal peptide. The results shown in this panel for ATP8B1 inhibition are the same as those displayed in panel (**B**) for a concentration of 180 µM C-terminal peptide. The rate of ATP hydrolysis was corrected for NADH photobleaching and the activity in the absence of the C-terminal peptide was taken as 100% for each species. **** $p < 0.0001$ according to two-way ANOVA with Tukey's test vs activity in the absence of peptide. ns: not significant. Data are a mean ± s.d. of three replicate experiments. Source files for (**B, C, D and E**) are available in *Figure 5—source data 1*, *Figure 5—source data 2*, *Figure 5—source data 3* and *Figure 5—source data 4*, respectively.

The online version of this article includes the following source data and figure supplement(s) for figure 5:

**Source data 1.** GraphPad Prism tables and curve fitting for results displayed in *Figure 5B*.

**Source data 2.** GraphPad Prism tables and curve fitting for results displayed in *Figure 5C*.

**Source data 3.** GraphPad Prism tables for results displayed in *Figure 5D*.

**Source data 4.** GraphPad Prism tables and statistical analysis for results displayed in *Figure 5E*.

**Figure supplement 1.** Effect of the ATP8B1 C-terminal peptide on the enzyme-coupled assay.

**Figure supplement 1—source data 1.** GraphPad Prism tables for results displayed in *Figure 5—figure supplement 1C*.

Under these conditions, PE and to a lesser extent PS, but not cardiolipin (CL) and sphingomyelin (SM) could stimulate ATP8B1 activity (*Figure 6A*). Plasma-membrane localized yeast P4-ATPases Dnf1 and Dnf2 have been shown to transport lyso-phosphatidylcholine (Lyso-PC) (*Riekhof et al., 2007*) and the alkylphosphocholine analogs miltefosine and edelfosine (*Hanson et al., 2003*), in addition to PC (*Pomorski et al., 2003*). Furthermore, when co-expressed with CDC50A, murine ATP8B1 was shown to increase uptake of the alkylphosphocholine analog perifosine in HeLa and HEK293T cells (*Munoz-Martínez et al., 2010*). As compared with background levels, Lyso-PC induced a clear increase in the ATP hydrolysis rate of ΔN42/C1174 ATP8B1. Weak activation was also observed in the presence of edelfosine and miltefosine (*Figure 6A*).

To further dissect the regulatory mechanism of ATP8B1-CDC50A, we examined the specificity of the purified enzyme for PPIns. All PPIn species were tested at the same molar concentration and at a fixed concentration of PC, and differed in the number and positions of phosphorylations on the inositol headgroup. Phosphorylation of the headgroup appeared to be essential for stimulating ATP8B1 ATPase activity, as no activity could be detected above background using phosphatidylinositol (*Figure 6B*). Monophosphorylated PPIn species, namely PI(3)P, PI(4)P, and PI(5)P, were equally efficient in stimulating ATP hydrolysis by ATP8B1. When the inositol ring was phosphorylated twice, the ATPase activity was increased about 2-fold compared to that observed with monophosphorylated PPIns (*Figure 6B*), with no dramatic difference in activity between PI(4,5)P$_2$, PI(3,4)P$_2$, and PI(3,5)P$_2$. Tri-phosphorylated PI(3,4,5)P$_3$ increased further the activity of ATP8B1 by about 1.5-fold. Thus, although the number of phosphorylations on the inositol ring matters, the positions do not and ATP8B1-CDC50A can be activated by a wide variety of PPIns with increasing efficiency linked

**Table 2.** Half-maximal inhibitory concentration (IC$_{50}$) values for the C-terminal peptide, in comparison with its phosphorylated form.

The values indicated in the table were deduced from dose-response curves displayed in *Figure 5B and C*. The number of data points used to calculate the IC$_{50}$ is indicated in parenthesis. IC$_{50}$ values are expressed as mean ± s.d.

| ATP8B1-CDC50A | Inhibitory peptide | IC$_{50}$ (µM) |
|---|---|---|
| ΔN42/C1174 (n = 33) | C-terminal | 22.1 ± 1.2 |
| ΔN42/C1174 (n = 35) | Phosphorylated C-terminal | 377.4 ± 227 |
| ΔC1174 (n = 34) | C-terminal | 0.081 ± 0.014 |
| ΔC1174 (n = 47) | Phosphorylated C-terminal | 1.96 ± 0.35 |

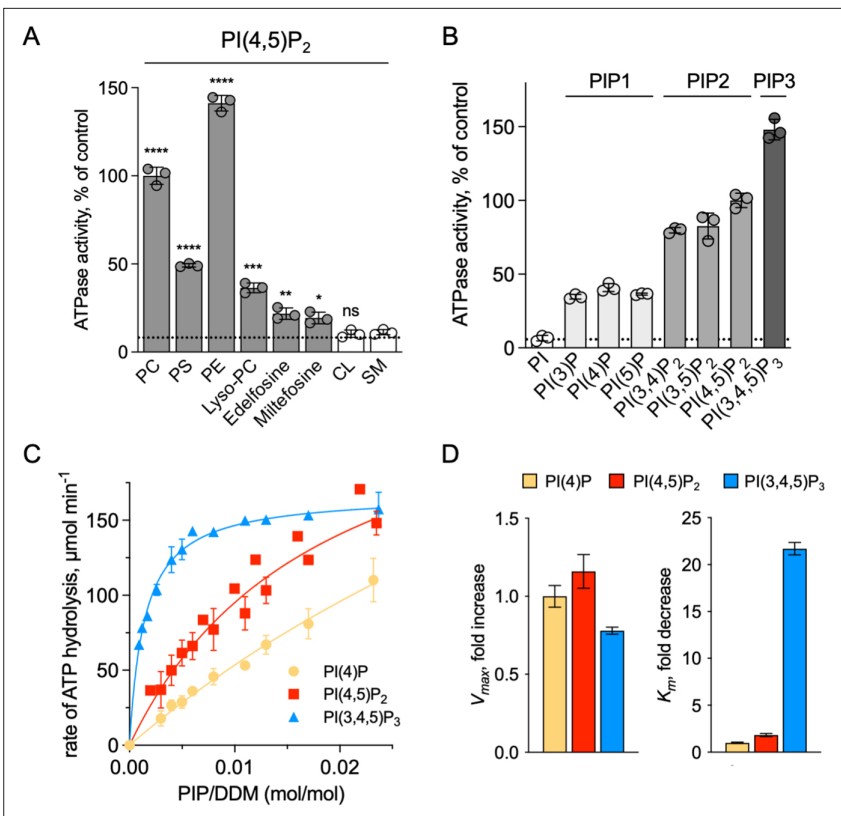

**Figure 6.** Sensitivity of ATP8B1-CDC50A to phospholipids. (**A**) ATPase activity of the ΔN42/C1174 ATP8B1 determined in the presence of various glycerophospholipids, lipid derivatives, and sphingomyelin, at 30 °C. The assay medium contained 1 mM MgATP, 1 mg ml$^{-1}$ DDM, and 0.01 mg ml$^{-1}$ CHS. PI(4,5)P$_2$ was added at 23 µM and the various lipids and lipid derivatives were added at 115 µM. The rate of ATP hydrolysis was corrected for NADH photobleaching occurring before the addition of the purified ATP8B1-CDC50A complex to the assay cuvette. The specific activity measured in the presence of PC and PI(4,5)P$_2$ was taken as 100% (~0.15–0.3 µmol min$^{-1}$ mg$^{-1}$). The dotted line represents the background activity measured in the absence of any added lipid. **** p < 0.0001, *** p = 0.0002, ** p = 0.0071, * p = 0.0177 according to unpaired two-tailed *t* test *vs* SM condition. ns: not significant. Data are mean ± s.d. of 3 replicate experiments. (**B**) ATPase activity of the ΔN42/C1174 ATP8B1 (0.5 µg ml$^{-1}$) determined in the presence of mono, di, and tri-phosphorylated phosphoinositides, at 30 °C. The activity was measured in the presence of 1 mg ml$^{-1}$ DDM, 0.1 mg ml$^{-1}$ CHS, 115 µM PC and 23 µM of the indicated phosphoinositides. The rate of ATP hydrolysis was corrected for NADH photobleaching occurring before the addition of the purified ATP8B1-CDC50A complex and ATP to the assay cuvette. The specific activity of the wild-type measured in the presence of PC and PI(4,5)P$_2$ was taken as 100%. The dotted line represents the activity measured in the sole presence of PC. Data are mean ± s.d. of three replicate experiments. (**C**) Apparent affinity of ΔN42/C1174 ATP8B1 ( ~ 3–3.3 µg ml$^{-1}$) for PI(4)P, PI(4,5)P$_2$ and PI(3,4,5)P$_3$. $K_m$ for phosphoinositides was measured at 37 °C in the presence of PC. The assay medium contained 1 mM MgATP, 0.5 mg ml$^{-1}$ DDM, 0.01 mg ml$^{-1}$ CHS, 23 µM PI(4,5)P$_2$ and 57 µM PC. Successive additions of DDM and PC gradually decreased the PI(4,5)P$_2$/DDM ratio. The PC/DDM ratio remained constant at 0.058 mol/mol. Plotted lines represent the best fit to a Michaelis-Menten equation. (**D**) Variations of the maximum velocity ($V_{max}$) and apparent affinity ($K_m$) of ΔN42/C1174 ATP8B1 for phosphoinositides calculated from double reciprocal plots displayed in *Figure 6—figure supplement 1*, with respect to that measured in the presence of PI(4)P. The data in (**C**) and (**D**) represent the mean ± s.d. of three to four replicate experiments. Source files for (**A, B, C and D**) are available in *Figure 6—source data 1*, *Figure 6—source data 2*, *Figure 6—source data 3* and *Figure 6—source data 4*, respectively.

The online version of this article includes the following source data and figure supplement(s) for figure 6:

**Source data 1.** GraphPad Prism tables and statistical analysis for results displayed in *Figure 6A*.

**Source data 2.** GraphPad Prism tables for results displayed in *Figure 6B*.

**Source data 3.** GraphPad Prism tables and curve fitting for results displayed in *Figure 6C*.

**Source data 4.** GraphPad Prism tables for results displayed in *Figure 6D*.

*Figure 6 continued on next page*

*Figure 6 continued*

**Figure supplement 1.** Determination of the kinetic parameters for activation of ATP8B1-CDC50A by PPIns.

**Figure supplement 1—source data 1.** GraphPad Prism tables and curve fitting for results displayed in *Figure 6—figure supplement 1*.

**Figure supplement 2.** Quantification of the detergent bound to the transmembrane domain of Drs2-Cdc50.

**Figure supplement 2—source data 1.** GraphPad Prism tables for results displayed in *Figure 6—figure supplement 2*.

to the number of phosphorylations. The differential activation by PPIns observed in *Figure 6B* could either be the result of a variation in the maximal velocity of ATP hydrolysis, the apparent affinity for PPIns, or both. To distinguish between these possibilities, we measured the rate of ATP hydrolysis by ATP8B1 in relation to the PPIn/detergent ratio (*Figure 6C*), taking PI(4)P, PI(4,5)$P_2$ and PI(3,4,5)$P_3$ as representative examples of singly, doubly and triply phosphorylated PPIns, respectively. Whereas double-reciprocal plots indicated comparable maximum ATP hydrolysis rates in the presence of PI(4)P, PI(4,5)$P_2$ and PI(3,4,5)$P_3$, the apparent affinity of ATP8B1 for PI(3,4,5)$P_3$ was found much higher than for PI(4)P and PI(4,5)$P_2$ (*Figure 6D*, *Figure 6—figure supplement 1*). Thus, ATP8B1 exhibits a strong preference for PI(3,4,5)$P_3$ over other PPIns in vitro.

## Discussion

Based on the cryo-EM structure of ATP8B1-CDC50A and dissection of its regulatory mechanism using biochemical assays, we identify the C-terminal extension of ATP8B1 as a central component in the regulation of its activity, and a cooperative contribution of the N-terminus of ATP8B1 in the auto-regulatory mechanism. Furthermore, we report that PPIns are essential activators of ATP8B1 activity and identify in vitro a preference for PI(3,4,5)$P_3$ in the activation of ATP8B1. Truncation of the C- and N-termini of ATP8B1 allows switching ATP8B1 from a fully inhibited to an activated form, provided lipid transport substrate and PPIns are present. Addition of a C-terminal peptide rescues inhibition, and inhibition is subject to regulation by phosphorylation at S1223 of the C-terminal extension.

### Autoinhibition of P4-ATPase flippases by their terminal tails: An evolutionarily conserved mechanism?

The autoinhibition of plasma membrane-localized ATP8B1 by its C-terminus is reminiscent of that observed for the yeast endomembrane homolog Drs2. While an intact Drs2-Cdc50 complex exhibits hardly any lipid-induced ATPase activity, once the C-terminus has been trimmed off by proteases, the complex becomes competent for ATP hydrolysis (*Azouaoui et al., 2017*). The ability of Drs2 to hydrolyze ATP requires not only displacement of its C-terminus but also the binding of PI(4)P (*Azouaoui et al., 2017*; *Timcenko et al., 2019*). Cryo-EM structures show that the C-terminus of Drs2 binds in a cleft between the P-domain and the N-domain, thus providing a structural explanation for autoinhibition (*Bai et al., 2019*; *Timcenko et al., 2019*). The C-terminus also appears to play a role in autoinhibition of ATP8A2, although this enzyme does not seem to be regulated by PPIns (*Chalat et al., 2017*). In particular, the C-terminus of ATP8A1, a close relative of ATP8A2, was recently shown to extend through its cytosolic catalytic domains (*Hiraizumi et al., 2019*). This raises the question as to whether such autoregulatory mechanism is a conserved feature among P4-ATPases. By comparing the sequences of P4-ATPase termini from various organisms (*Figure 7A*, *Figure 4—figure supplement 1* for a full alignment), it appears that although the C-termini of P4-ATPases are in general poorly conserved, one exception to this rule is the ATP8B1 AYAFS motif which occupies the ATP-binding site. Furthermore, in the autoinhibited Drs2 and ATP8A1 structures, their C-termini overlap extensively despite a rather low sequence conservation (*Figure 7B*). Noteworthy, the C-terminal peptide of ATP8B1 did not exhibit an inhibitory effect on Drs2 (*Figure 5E*), suggesting that autoinhibition per se is mainly driven by the region downstream the conserved motif, the latter mediating the interaction between the A and N domain. Thus, we predict that any P4-ATPase containing the (G/A)(Y/F)AFS motif is likely to be autoinhibited by its C-terminus. We further propose that autoinhibition might be occurring in a conformation-dependent manner. Indeed, previous structural work from Hiraizumi and colleagues, capturing an almost complete catalytic cycle of full-length ATP8A1-CDC50A, showed

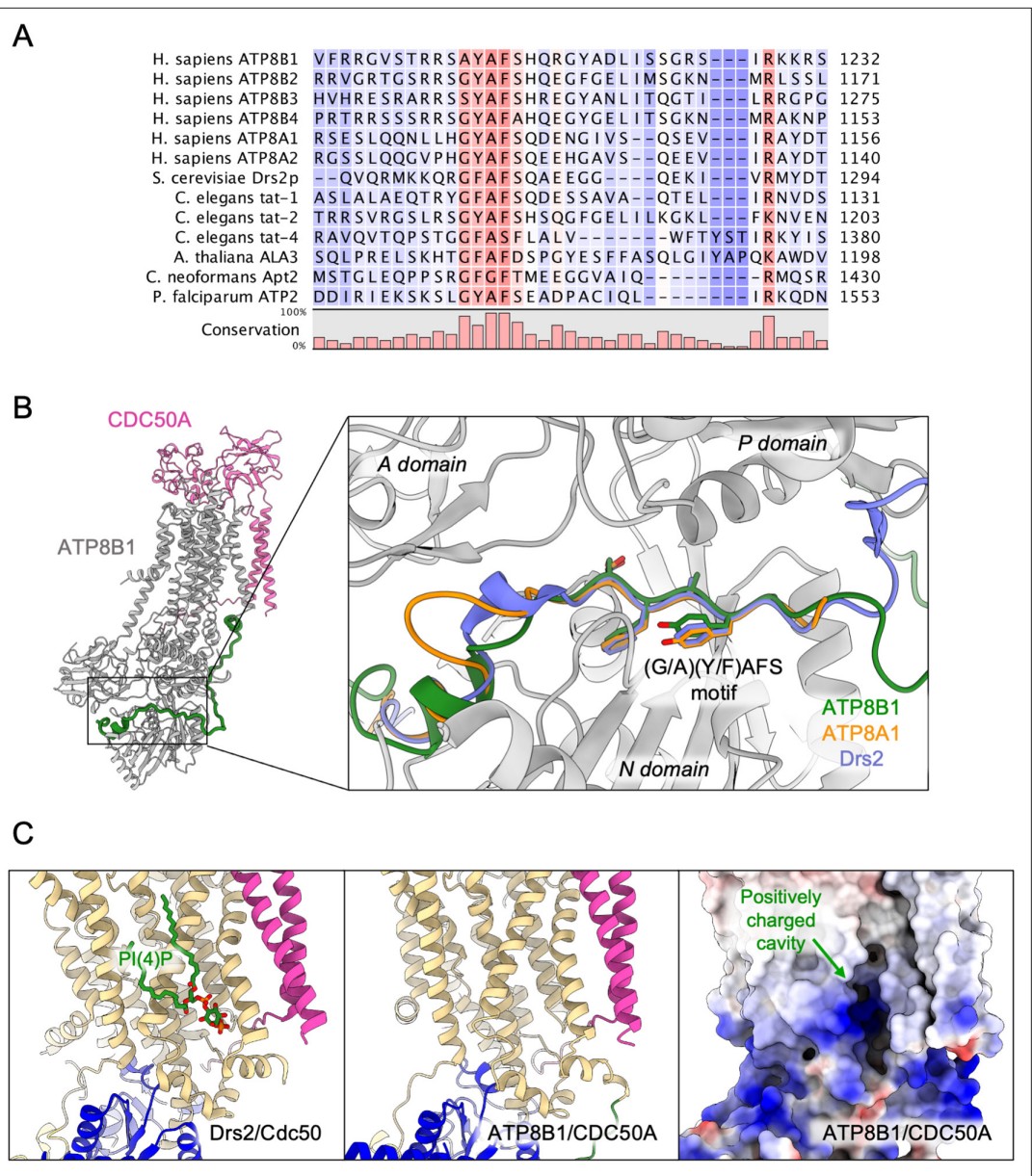

**Figure 7.** Proposed mechanism for autoinhibition and regulation by phosphoinositides of the ATP8B1-CDC50A complex. (**A**) Sequence alignment of select P4-ATPases C-termini, including ATP8B1, ATP8A1 and Drs2, which are all known to be autoinhibited. The shading indicates conservation (blue 0% – red 100%). (**B**) Comparison of the binding sites of ATP8B1, ATP8A1 (PDB ID: 6K7N) and Drs2 (PDB ID: 6ROH) C-terminal tails, respectively in green, orange and blue reveals a common architecture and location of the inhibitory C-termini, and specifically the conserved (G/A)(Y/F)AFS motif (AYAFS for ATP8B1, GYAFS for ATP8A1 and GFAFS for Drs2) located in the ATP binding pocket. (**C**) Side view of the PI(4)P-binding site of Drs2 (left). PI(4)P (in stick representation) is bound in the membrane domain. The same region in ATP8B1 reveals a similar organization (middle) with the presence of a positively-charged cavity (right) suggesting a putative phosphoinositide binding pocket in ATP8B1. CDC50A and Cdc50 transmembrane helices are colored in pink.

The online version of this article includes the following figure supplement(s) for figure 7:

**Figure supplement 1.** Structural comparison of ATP8B1 and Ypk9 autoinhibition mechanism.

that the inhibitory C-terminus is observed only in the BeF$_x$-stabilized E2P form and is completely disordered in other conformations, suggesting that autoinhibition specifically occurs in the E2P state. We also show in *Figure 1D* that full-length ATP8B1 may be phosphorylated from [γ-$^{32}$P]ATP, indicating that in the E1 state, the presence of the C-terminal tail does not prevent accessibility of the

nucleotide-binding site. As such, we foresee that the C-terminal tail is in equilibrium between a state bound to the ATP8B1 cytosolic domains and an unbound state, this equilibrium being poised toward the bound state in the E2P conformation.

Our study also identifies a previously unrecognized role for the N-terminal tail of ATP8B1 in the autoinhibition process. Although the precise mechanism is so far uncertain, our data indicate that the N-terminal tail of ATP8B1 has a strong synergistic effect on the autoinhibition by its C-terminal extension (*Figures 4B and 5B–D*). Owing to numerous interactions observed in our structure of ATP8B1, the N-terminal tail might restrain the flexibility of the A-, N-, and P-domains necessary for nucleotide binding to the N-domain and catalysis, even in the absence of the C-terminal tail. Another non-exclusive possibility could be that the N-terminal tail prevents dissociation of the C-terminus by locking down the N-domain through electrostatic interaction with S598. A functional cooperation between N- and C-termini has previously been described for the plant H⁺-ATPase, a P-type ATPase from the P3 subfamily, where modifications in the N-terminus result in kinase-mediated phosphorylation in the C-terminus, eventually leading to activation of the pump (*Ekberg et al., 2010*). Moreover, recent cryo-EM structures revealed an autoinhibitory role for the N-terminus of the P5B-ATPase Ypk9 mediated by its interaction with the cytosolic domains (*Figure 7—figure supplement 1*), and it was proposed in this study that the C-terminal tail of Ypk9 may also play a functional role owing to its interaction with the P-domain (*Li et al., 2021*).

## Phosphorylation as a mechanism for the regulation of ATP8B1 activity

The inhibitory properties of a peptide derived from the C-terminus of ATP8B1 suggest that phosphorylation of residue S1223 plays an important role. Identification of the corresponding residue (S1223) from the mouse orthologue ATP8B1 in large-scale phosphoproteomic studies (*Huttlin et al., 2010*; *Villén et al., 2007*), suggests that phosphorylation of S1223 in human ATP8B1 might be part of the activating mechanism that lifts autoinhibition in vivo. Consistent with this hypothesis, calcium/calmodulin-dependent protein kinase II (CaMKII) has been shown to phosphorylate a serine residue, S1138, in the autoinhibitory C-terminus of bovine ATP8A2. Substitution of S1138 to alanine resulted in a 33% loss of the PS-dependent ATPase activity of ATP8A2 (*Chalat et al., 2017*). Canalicular transporters also involved in inherited forms of intrahepatic cholestasis such as the bile salt export pump (PFIC2, ABC11B) have been found phosphorylated by PKC when overexpressed in insect cells (*Noe et al., 2001*) and the floppase ABCB4 (PFIC3), known to transport PC in the opposite direction compared to ATP8B1, was shown to be stimulated by PKA- and PKC-dependent phosphorylation (*Gautherot et al., 2014*). Future studies are required to identify kinases responsible for the phosphorylation of S1223 and other sites, to investigate the functional consequences of ATP8B1 phosphorylation on its activity, both in vitro and in vivo.

## Regulation of ATP8B1-CDC50A by Phosphoinositides

In this study, we identified PPIns as regulators of ATP8B1 ATPase activity. It must be pointed out that the activity of the intact full-length ATP8B1 is not stimulated by addition of PI(4,5)P$_2$ (*Figure 4B*) and that the C-terminus of ATP8B1 must be removed for PI(4,5)P$_2$ to exert its stimulatory effect. While it remains possible that phosphoinositides participate in autoinhibition relief, as proposed for the yeast Drs2-Cdc50 flippase complex, this suggests that phosphoinositides mediate their activatory effect through a distinct mechanism that does not involve the tails, for example by promoting conformational changes in the membrane domain that could for instance regulate access to the substrate-binding site. Whereas all PPIns showed the ability to stimulate ATP8B1 activity (*Figure 6*), PI(3,4,5)P$_3$ displayed a much higher affinity for ATP8B1 than other PPIns. The $K_m$ value for activation of ATP8B1 by PI(3,4,5)P$_3$ is about $1.4.10^{-3}$ mol PI(3,4,5)P$_3$/mol DDM. Based on our own estimation of the number of DDM molecules surrounding the transmembrane domain of Drs2-Cdc50 using size-exclusion chromatography in the presence of $^{14}$C-labeled DDM (*Figure 6—figure supplement 2*), we estimate that the detergent micelle around the transmembrane region of ATP8B1-CDC50A is composed of ~270 ± 56 molecules of DDM. Taking into account the additional presence of two transmembrane helices contributed by Cdc50, this is the same order of magnitude as the amount of DDM bound to purified SERCA1a (155 ± 27 mol DDM/mol SERCA1a), a P-type ATPase from the P2 subfamily, as determined by MALDI-TOF mass spectrometry (*Chaptal et al., 2017*). A $K_m$ value of $1.4 \times 10^{-3}$ mol PI(3,4,5)P$_3$/mol DDM corresponds to ~0.38 mol of PI(3,4,5)P$_3$ per 270 mol of DDM (or 0.14 mol%) in the immediate

environment of ATP8B1-CDC50A, emphasizing the strong affinity of ATP8B1 for PI(3,4,5)P$_3$. This is consistent with PPIns being activators rather than substrates as is the case for PI(4)P towards the yeast Drs2-Cdc50 complex. PI(3,4,5)P$_3$ is primarily localized at the plasma membrane, and one of the least abundant PPIns in mammalian cells, being virtually undetectable in quiescent cells. The tight control of PI(3,4,5)P$_3$ concentration stems from its critical role in key signalling pathways such as cell proliferation, survival and membrane trafficking (*Marat and Haucke, 2016*). Interestingly, a recent report provided quantitative analysis of phosphoinositides, including PI(3,4,5)P$_3$, in the plasma membrane of MT-4 cells, a T-lymphocyte cell line. In these cells, PI(3,4,5)P$_3$ represents 0.00025% of total plasma membrane lipids (*Mücksch et al., 2019*). However, upon activation of cell-surface receptors and recruitment of class I PI3-kinases, PI(3,4,5)P$_3$ levels may rise up to 100-fold (*Clark et al., 2011*), suggesting that its concentration may rise up to 0.025 mol% in the PM. Although comparison must be made with care, due to the fact that activation of ATP8B1 by PI(3,4,5)P$_3$ may be different in lipid bilayers and solubilized systems, it is worth noting that 0.025 mol% of PI(3,4,5)P$_3$ in the PM is in the same range as 0.14 mol%, the PI(3,4,5)P$_3$ concentration required to reach half-maximal activity of ATP8B1 in detergent micelles.

The lower $K_m$ of ATP8B1 for PI(3,4,5)P$_3$ than for other PPIns suggests that the cavity where PI(3,4,5)P$_3$ binds is specifically adjusted to this PPIn, whereas other PPIns can fit as well, but less efficiently. To our knowledge, direct regulation of integral membrane proteins by PI(3,4,5)P$_3$ has not previously been shown. Intriguingly, despite addition of PI(3,4,5)P$_3$ during sample preparation for cryo-EM studies, no clear density could be observed for this lipid. However, the cavity lined by TM7, TM8, and TM10 on the structure of ATP8B1, which corresponds to the PI(4)P binding site in Drs2, consists of a large number of basic residues (*Figure 7C*) strongly hinting at a similar site in both Drs2 and ATP8B1. On the other hand, the role of PPIns on the activation of ATP8B1 with C-terminal or double N- and C-terminal truncation could be interpreted as supporting a model where regulatory PPIns bind to the N-terminal tail of ATP8B1. Interestingly, the N-terminal tail of ATP8B1 contains a patch of positively charged residues between P42 and D70 (including R46, R49, R55, R59 and K60), a region which is not visible in our structure. This would be reminiscent of the proposed model for the P5-ATPase ATP13A2, where binding of the negatively charged lipids phosphatidic acid and PI(3,5)P$_2$ to the N-terminal domain stimulates catalytic activity (*Holemans et al., 2015*; *Tomita et al., 2021*).

Irrespective of this, the physiologically relevant regulatory PPIn is still unknown. Given the localization of ATP8B1 in the apical membrane of epithelial cells in mammals, and the subcellular localization and abundance of PPIns in cell membranes (*Balla, 2013*; *Dickson and Hille, 2019*), both PI(3,4,5)P$_3$ and PI(4,5)P$_2$ might fulfill this task. Future studies aimed at manipulating PPIns levels in living cells should help reveal whether ATP8B1 depends on specific PPIns in vivo, opening the way to modulate functional levels of ATP8B1 in cells.

## Structural basis for catalytic deficiency induced by inherited ATP8B1 mutations

Our structural model of ATP8B1 enabled us to map the mutations found in patients suffering from PFIC1, BRIC1, or ICP1 (*Bull et al., 1998*; *Deng et al., 2012*; *Dixon et al., 2017*; *Klomp et al., 2004*; *Painter et al., 2005*; *Figure 8A*). Mutations are homogenously distributed along the protein sequence, and some mutations are likely to impair catalytic properties of ATP8B1 directly (*Figure 8B*). Mutations D554N and H535L are located in the nucleotide-binding pocket, suggesting that these mutations might prevent or affect ATP binding. The D554 residue is at interacting distance with the autoinhibitory C-terminus and its mutation might also alter autoregulation. Additionally, mutations S453Y, D454G, and T456M in the P-domain will abolish autophosphorylation of the catalytic aspartate (D454), thus resulting in an inactive ATP8B1.

The structure of ATP8B1 presented in this report is locked in a E2P$_{autoinhibited}$ state where the exoplasmic lipid pathway is closed. However, it is important to note that numerous mutations can be found in this region (*Figure 8B*). In particular, the S403 residue, mutated to a tyrosine in PFIC1, is part of the PISL motif conserved in P4-ATPases. The PISL motif is located in TM4 and has been shown to interact with the phosphoglycerol backbone of PS, the transport substrate of Drs2 and ATP8A1 (*Hiraizumi et al., 2019*; *Timcenko et al., 2021*). A relatively conservative mutation of this Ser into Ala in ATP8A2 (S365A), has been shown to significantly diminish its ATPase activity and transport substrate affinity (*Vestergaard et al., 2014*). Moreover, mutations E981K and L127P have also been shown to

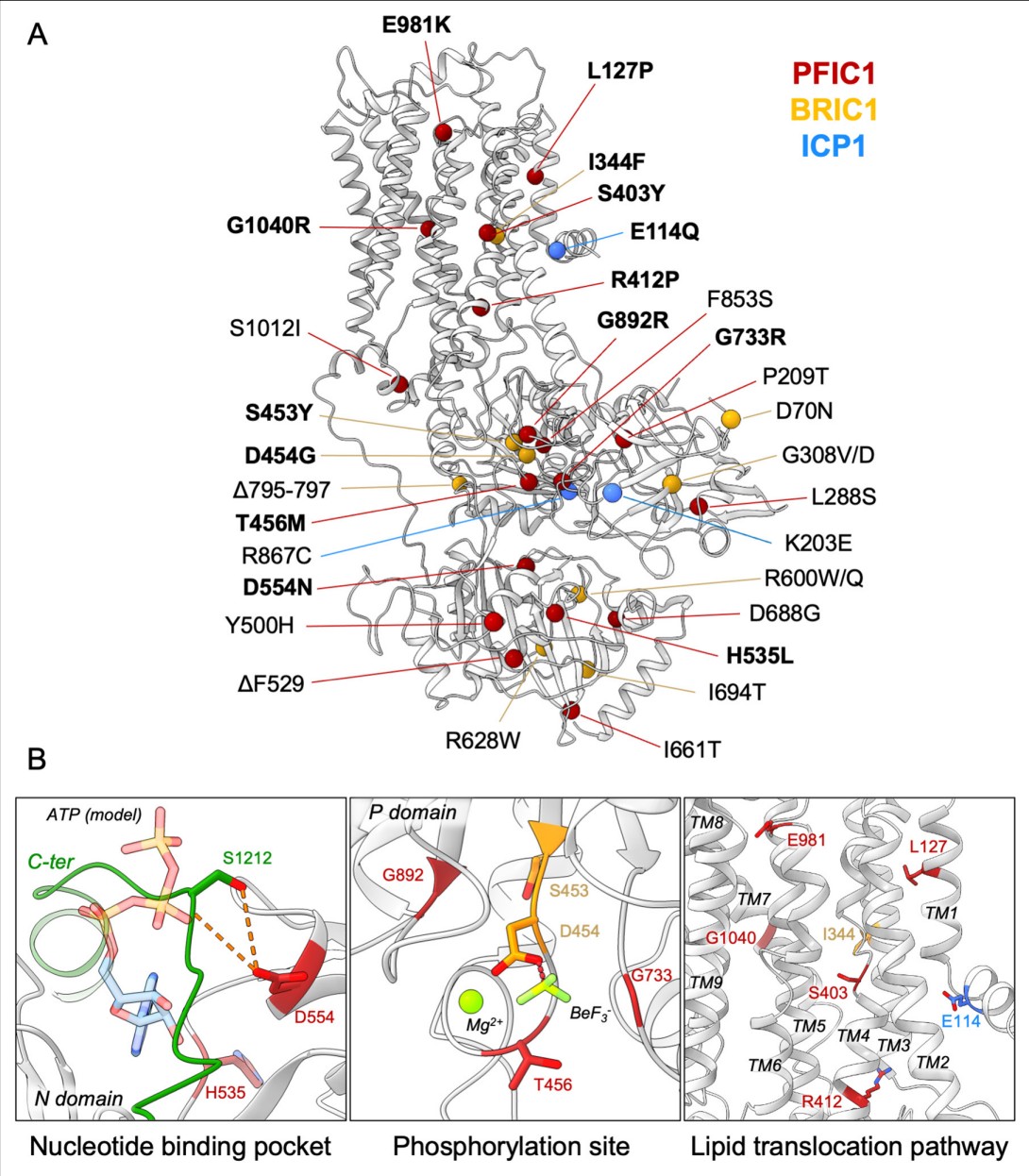

**Figure 8.** Structural map of the inherited intrahepatic cholestasis-related mutations. (**A**) Mutations found in PFIC1, BRIC1 or ICP1 patients are respectively shown as red, yellow and blue spheres on ATP8B1 E2P$_{autoinhibited}$ structure (in grey). Mutations indicated in bold are presented in panel (**B**). (**B**) Close-up views of the nucleotide binding site within the N-domain of ATP8B1. The ATP molecule position was modeled by aligning ATP8B1 N-domain with the N-domain of ATP8A1 in E1-ATP bound state (PDB: 6K7J) (left). (Middle) the phosphorylation site in the P-domain with Mg$^{2+}$ and the phosphate mimic BeF$_3^-$ in green. (Right) the lipid transport pathway.

impair ATP8B1-catalyzed transport of PC in vivo (*Takatsu et al., 2014*). Mutation of the corresponding residues in the PS-specific ATP8A2 alters ATPase activity and lipid specificity (*Gantzel et al., 2017*). Further functional and structural studies will be needed to better understand how these mutations may affect substrate recognition and translocation.

## Conclusions

Our findings show that the plasma membrane P4-ATPase ATP8B1 is tightly regulated by its N- and C-terminal tails as well as PPIns and that the autoinhibitory mechanism can be mimicked by exogenous peptides. Understanding the regulatory mechanism of mammalian P4-ATPases will be instrumental for the subsequent design of molecules that would enforce/mimic or stimulate the release of

the autoinhibitory C-terminus. We propose that the regulatory mechanism uncovered in this study may be a feature shared by other P4-ATPases, and that phosphorylation of the C-terminal tail of ATP8B1 is likely to be involved in the regulation of ATP8B1 activity. Moreover, these studies will pave the way towards detailed functional assessment of disease-associated ATP8B1 mutations found in PFIC1 patients and towards the design of both activating and inhibiting compounds of P4-ATPases, based on regulatory mechanisms in vivo.

# Materials and methods

## Key resources table

| Reagent type (species) or resource | Designation | Source or reference | Identifiers | Additional information |
|---|---|---|---|---|
| Strain, strain background (*Saccharomyces cerevisiae*, MATα) | W303.1b/ *Δpep4* | López-Marqués laboratory | | Strain deficient for the main vacuolar protease |
| Strain, strain background (*Saccharomyces cerevisiae*, MATα) | W303.1b/ *GAL4-2* | Pompon Laboratory | | Additional copy of the *GAL4* gene in the yeast chromosome |
| Antibody | FIC1 (H-91) rabbit polyclonal anti-ATP8B1 antibody | Santa-Cruz Biotechnology | Cat#sc-134967 | (1:10000) This product has been discontinued |
| Antibody | Goat anti-rabbit HRP-coupled polyclonal IgG antibody | Biorad | Cat#1706515 | (1:2000) |
| Recombinant DNA reagent | ATP8B1 cDNA | Joost Holthuis laboratory | Uniprot: O43520 | |
| Recombinant DNA reagent | CDC50 cDNA | Joost Holthuis laboratory | Uniprot: Q9NV96 | |
| Peptide, recombinant protein | ATP8B1 C-terminal peptide | Biomatik Company | | |
| Peptide, recombinant protein | ATP8B1 phosphorylated C-terminal peptide | Biomatik Company | | Phosphorylated on S1223 |
| Peptide, recombinant protein | HRV 3 C protease | This study | | Expressed (pGEX-4T-2) and purified in Lenoir laboratory. The purification procedure of N-terminally tagged HRV 3 C protease can be found in the Materials and methods section |
| Peptide, recombinant protein | TEV protease | This study | | Expressed (pRK793) and purified in Lenoir laboratory. The purification procedure of N-terminally tagged TEV protease can be found in the Materials and methods section |
| Commercial assay or kit | NucleoSpin Plasmid, Mini kit for Plasmid DNA | Macherey-Nagel | Cat#740588.250 | |
| Commercial assay or kit | QuickChange II XL site-directed mutagenesis kit | Agilent technologies | Cat#200,521 | |
| Commercial assay or kit | Amicon 100 kDa cutoff | EMD Millipore | Cat#UFC510024 | For volume ≤0.5 ml |
| Commercial assay or kit | Vivaspin 500 | Sartorius | Cat#VS0142 | For volumes from 0.5 to 0.005 ml |
| Commercial assay or kit | Vivaspin 6 | Sartorius | Cat#VS0641 | For volumes from 0.5 to 6 ml |
| Commercial assay or kit | Vivaspin 20 | Sartorius | Cat#VS2041 | For volumes from 2 to 20 ml |

*Continued on next page*

*Continued*

| Reagent type (species) or resource | Designation | Source or reference | Identifiers | Additional information |
|---|---|---|---|---|
| Commercial assay or kit | Superose 6 Increase 10/300 GL | GE Healthcare/ Cytiva | Cat#29091596 | |
| Commercial assay or kit | TSK3000-SW | Tosoh Bioscience | Cat#08541 | |
| Commercial assay or kit | Streptavidin-sepharose resin | GE Healthcare/ Cytiva | Cat#17511301 | |
| Chemical compound, drug | *n*-dodecyl-β-D-maltopyranoside, Anagrade | Anatrace | Cat#D310 | |
| Chemical compound, drug | Cholesteryl hemisuccinate | Sigma | Cat#C6013 | |
| Chemical compound, drug | Lauryl maltose neopentyl glycol | Anatrace | Cat#NG310 | |
| Chemical compound, drug | Sodium chloride | ROTH | Cat#3957.2 | |
| Chemical compound, drug | Potassium chloride | Sigma-Aldrich | Cat#P9541 | |
| Chemical compound, drug | Magnesium chloride | Sigma-Aldrich | Cat#M2670 | |
| Chemical compound, drug | MOPS | Sigma-Aldrich | Cat#M1254 | |
| Chemical compound, drug | ATP | Sigma-Aldrich | Cat#A2383 | |
| Chemical compound, drug | Phospho(enol)pyruvic acid | Sigma-Aldrich | Cat#860,077 | |
| Chemical compound, drug | β-nicotinamide adenine dinucleotide, reduced disodium salt hydrate (NADH) Grade I, disodium salt | Roche | Cat#10107730001 | |
| Chemical compound, drug | Glycerol | VWR Chemicals | Cat#24387.292 | |
| Chemical compound, drug | D-glucose | Becton Dickinson | Cat#215,530 | |
| Chemical compound, drug | D-galactose | Sigma Aldrich | Cat#G5388 | |
| Chemical compound, drug | SIGMAFAST EDTA-free protease inhibitor cocktail | Sigma | Cat#S8830 | |
| Chemical compound, drug | Brain phosphatidylinositol-4-phosphate (PI4P) | Avanti Polar Lipids, Inc | Cat#840045 P | |
| Chemical compound, drug | Brain phosphatidylinositol-4,5-bisphosphate (PI(4,5)P$_2$) | Avanti Polar Lipids, Inc | Cat#840046 P | |
| Chemical compound, drug | 1,2-dioleoyl-*sn*-glycero-3-phospho-(1'-myo-inositol-3'-phosphate) (PI(3)P) | Avanti Polar Lipids, Inc | Cat#850150 P | |
| Chemical compound, drug | 1,2-dioleoyl-*sn*-glycero-3-phospho-(1'-myo-inositol-5'-phosphate) (PI(5)P) | Avanti Polar Lipids, Inc | Cat#850152 P | |

*Continued on next page*

*Continued*

| Reagent type (species) or resource | Designation | Source or reference | Identifiers | Additional information |
|---|---|---|---|---|
| Chemical compound, drug | 1,2-dioleoyl-*sn*-glycero-3-phospho-(1'-myo-inositol-3',4'-bisphosphate) (PI(3,4)P$_2$) | Avanti Polar Lipids, Inc | Cat#850153 P | |
| Chemical compound, drug | 1,2-dioleoyl-*sn*-glycero-3-phospho-(1'-myo-inositol-3',5'-bisphosphate) (PI(3,5)P$_2$) | Avanti Polar Lipids, Inc | Cat#850154 P | |
| Chemical compound, drug | 1,2-dioleoyl-*sn*-glycero-3-phospho-(1'-myo-inositol-3',4',5'-trisphosphate) (PI(3,4,5)P$_3$) | Avanti Polar Lipids, Inc | Cat#850156 P | |
| Chemical compound, drug | Brain phosphatidylserine (PS) | Avanti Polar Lipids, Inc | Cat#840032 P | |
| Chemical compound, drug | 1-palmitoyl-2-oleoyl-*sn*-glycero-3-phosphocholine (POPC) | Avanti Polar Lipids, Inc | Cat#850457 P | |
| Chemical compound, drug | 1-palmitoyl-2-oleoyl-*sn*-glycero-3-phosphoethanolamine (POPE) | Avanti Polar Lipids, Inc | Cat#850757 P | |
| Chemical compound, drug | 1-palmitoyl-2-oleoyl-*sn*-glycero-3-phosphoserine (POPS) | Avanti Polar Lipids, Inc | Cat#840034 P | |
| Chemical compound, drug | Bovine heart cardiolipin (CL) | Avanti Polar Lipids, Inc | Cat#840012 P | |
| Chemical compound, drug | egg chicken sphingomyelin (SM) | Avanti Polar Lipids, Inc | Cat#860061 P | |
| Chemical compound, drug | edelfosine | Avanti Polar Lipids, Inc | Cat#999995 P | |
| Chemical compound, drug | Miltefosine (Fos-Choline-16) | Anatrace | Cat#F316 | |
| Chemical compound, drug | 1-stearoyl-2-hydroxy-sn-glycero-3-phosphocholine (Lyso-PC) | Sigma | Cat#L2131 | |
| Chemical compound, drug | Pyruvate kinase | Sigma | Cat#P7768 | |
| Chemical compound, drug | Lactate dehydrogenase | Sigma | Cat#L1006 | |
| Chemical compound, drug | [γ-$^{32}$P]ATP | Perkin-Elmer | Cat#BLU002A | |
| Chemical compound, drug | His-probe-HRP | Thermo Scientific | Cat#15,165 | |
| Software, algorithm | EPU v 2.3 | Thermo Fisher | https://www.thermofisher.com/it/en/home/electron-microscopy/products/software-em-3d-vis/epu-software.html | |
| Software, algorithm | cryoSPARC v3 | *Punjani et al., 2017*, Structura Biotechnology Inc | https://www.nature.com/articles/nmeth.4169 | |

*Continued*

| Reagent type (species) or resource | Designation | Source or reference | Identifiers | Additional information |
|---|---|---|---|---|
| Software, algorithm | ChimeraX 1.4 | *Goddard et al., 2018* | https://www.cgl.ucsf.edu/chimerax/ | |
| Software, algorithm | I-TASSER | *Yang et al., 2015* | https://zhanggroup.org/I-TASSER/ | |
| Software, algorithm | Coot 0.9.6 | *Emsley et al., 2010* | https://doi.org/10.1107/S0907444904019158 https://www2.mrc-lmb.cam.ac.uk/personal/pemsley/coot/ | |
| Software, algorithm | Phenix 1.19.2 | *Liebschner et al., 2019* | https://doi.org/10.1107/S2059798318006551 http://phenix-online.org/ | |
| Software, algorithm | Molprobity 4.5.1 | *Williams et al., 2018* | https://doi.org/10.1002/pro.3330 http://molprobity.biochem.duke.edu | |
| Software, algorithm | ImageJ | *Schneider et al., 2012* | https://imagej.nih.gov/ij/ | |
| Software, algorithm | Prism 9 | GraphPad | https://www.graphpad.com/scientific-software/prism/ | |
| Other | C-Flat 1.2/1.3 Cryo-EM Grid - Copper (400 Grid Mesh, 20 nm Carbon Thickness) | Molecular Dimensions | CF-1.2/1.3-4CU-50 | Support film for biological samples in cryo-EM techniques -20 nm C-flat carbon film with 1.2 μm hole size and 1.3 μm hole spacing |

## Materials

Products for yeast and bacteria cultures were purchased from Difco (BD Biosciences) and Sigma. DNA Polymerase, restriction and modification enzymes, as well as Endoglycosidase H-MBP, were purchased from New England Biolabs (NEB). Lauryl Maltose Neopentyl Glycol (LMNG, NG310), *n*-dodecyl-β-D-maltopyranoside (DDM, D310) and miltefosine (also known as Fos-choline-16, FC-16, F316) were purchased from Anatrace. Cholesteryl hemisuccinate (CHS, C6013) and 1-stearoyl-2-hydroxy-sn-glycero-3-phosphocholine (Lyso-PC) were purchased from Sigma. Brain phosphatidylinositol-4-phosphate (PI(4)P), brain phosphatidylinositol-4,5-bisphosphate (PI(4,5)P$_2$), 1,2-dioleoyl-*sn*-glycero-3-phospho-(1'-myo-inositol-3'-phosphate) (PI(3)P), 1,2-dioleoyl-*sn*-glycero-3-phospho-(1'-myo-inositol-5'-phosphate) (PI(5)P), 1,2-dioleoyl-*sn*-glycero-3-phospho-(1'-myo-inositol-3',4'-bisphosphate) (PI(3,4)P$_2$), 1,2-dioleoyl-*sn*-glycero-3-phospho-(1'-myo-inositol-3',5'-bisphosphate) (PI(3,5)P$_2$), 1,2-dioleoyl-*sn*-glycero-3-phospho-(1'-myo-inositol-3',4',5'-trisphosphate) (PI(3,4,5)P$_3$), brain phosphatidylserine (PS), 1-palmitoyl-2-oleoyl-*sn*-glycero-3-phosphocholine (POPC), 1-palmitoyl-2-oleoyl-*sn*-glycero-3-phosphoethanolamine (POPE), 1-palmitoyl-2-oleoyl-*sn*-glycero-3-phosphoserine (POPS), heart cardiolipin (CL), egg sphingomyelin (SM) and edelfosine were purchased from Avanti Polar lipids. The ATP8B1 C-terminal peptide RRSAYAFSHQRGYADLISSGRSIRKKRSPLDAIVADGTAEYRRTGDS, encompassing residues 1205–1251, and its S1223 phosphorylated derivative, were ordered from Biomatik Company (Biomatik, Ontario, Canada). Both peptides were resuspended at 1 mM in 50 mM MOPS-Tris pH 7, 100 mM KCl, 1 mM dithiothreitol (DTT). ATP8B1 was detected using a mouse anti-ATP8B1 antibody from Santa Cruz Biotechnology (Epitope: 1161–1251, ref: SC-134967, no longer available). An anti-rabbit HRP-coupled antibody (1706515) was purchased from Biorad. His-tagged CDC50A was detected using a His-probe -HRP from Thermo Scientific (15165). Precast stain-free gradient gels for tryptophan fluorescence (4568084) as well as Precision Plus Protein Standards (1610393) were purchased from Biorad. Pyruvate kinase (P7768), lactate dehydrogenase (L1006), and an EDTA-free protease inhibitor cocktail (S8830) were purchased from Sigma. [γ-$^{32}$P]ATP was purchased from Perkin-Elmer (BLU002A). Streptavidin-sepharose resin was purchased from GE/Cytiva (17511301).

**Table 3.** Primers used in this study.

| Primers | |
|---|---|
| FwBad ATP8B1 | 5'- ACAGTTTAAACGGTGGTGAGAATCTTTATTTT CAGGGCGGTGGTGGTGGTATGAGTACAGAAAGAGACTCAG - 3' |
| RevBad ATP8B1 | 5'- AGCATGGAGCTCTCAGCTGTCCCCGGTGCGCCTGTA - 3' |
| FwHis CDC50A | 5' – CACAGAATTCTAGTATGCATCATCATCATCATCATCAT CATCATCACCTAGGTGGTATGGCGATGAACTATAACGCG – 3' |
| RevHis CDC50A | 5' – CACAGAGCTCCTAAATGGTAATGTCAGCTGTATTAC - 3' |
| Fwd D454N | 5'- GATCCATTATATCTTCTCTAATAAGACGGGGACACTCACAC –3' |
| Rev D454N | 5'- GTGTGAGTGTCCCCGTCTTATTAGAGAAGATATAATGGATC –3' |
| Fwd 3 C-P43 | 5' – CTGGAGGTGCTGTTCCAGGGCCCGG AACAAAACCGAGTCAACAGGGAAGC – 3' |
| Rev 3 C-P43 | 5' – CGGGCCCTGGAACAGCACCTCCAGTG GTTCAACAGCAGACCCCTGGTCATCAAG – 3' |
| Fwd 3C-E1174 | 5' – CTGGAGGTGCTGTTCCAGGGCCCGAGTGATAAGATCCAGAAGCATC – 3' |
| Rev 3C-E1174 | 5' – CGGGCCCTGGAACAGCACCTCCAGTTCTGATGGCCAGATGGTCAT– 3' |

The pig kidney **α1β1** isoform of Na$^+$/K$^+$-ATPase was a kind gift from Natalya U. Fedosova, and microsomal membranes were prepared as previously described (***Klodos et al., 2002***).

## Yeast strains and plasmids

The *Saccharomyces cerevisiae* W303.1b/*Δpep4* (*MATα, leu2-3, his3-11, ura3-1, ade2-1, Δpep4, can$^r$, cir$^+$*) yeast strain was used for co-expression of ATP8B1 and CDC50A. The cDNAs encoding human ATP8B1 (hATP8B1, Uniprot: O43520; A1154T natural variant) and human CDC50A (hCDC50A, Uniprot: Q9NV96) were a kind gift from Joost Holthuis (University of Osnabruck, Germany). hATP8B1 was supplemented at its 5' end with a sequence coding a biotin acceptor domain (BAD), and a sequence coding a TEV protease cleavage site. The cleavage site was flanked by 2 glycines toward BAD and 4 glycines toward hATP8B1. Similarly, a sequence coding a decahistidine tag was added at the 5' end of hCDC50A. The tagged genes were cloned in a unique co-expression pYeDP60 plasmid (***Jacquot et al., 2012***). In this vector, hATP8B1 and hCDC50A are both placed under the control of a strong galactose-inducible promoter, *GAL10/CYC1*. The D454N mutation was introduced by site-directed mutagenesis using the QuickChange II XL site-directed mutagenesis kit (Agilent technologies). An

**Table 4.** Plasmids used in this study.

| Plasmids | References |
|---|---|
| pYeDP60_BAD-TevS-ATP8B1 (WT) / His$_{10}$CDC50A | This study |
| pYeDP60_BAD-TevS-ATP8B1 (D454N) / His$_{10}$CDC50A | This study |
| pYeDP60_BAD-TevS-ATP8B1 (P42-3CS) / His$_{10}$CDC50A | This study |
| pYeDP60_BAD-TevS-ATP8B1 (P42-3CS) / His$_{10}$CDC50A | This study |
| pYeDP60_BAD-TevS-ATP8B1 (E1174-3CS) / His$_{10}$CDC50A | This study |
| pYeDP60_BAD-TevS-ATP8B1 (P43 +E1174-3 CS) / His$_{10}$CDC50A | This study |
| pRK793 MBP-Tev$_{site}$-His$_7$-TEV$_{S219V}$-Arg$_5$ | ***Kapust et al., 2001*** |
| pGEX-4T-2 His$_6$-Arg$_8$-GST-3C | |

overlap extension PCR strategy was used to insert the 3 C protease site (LEVLFQGP) between Pro42 and Glu43 and/or between Glu1174 and Ser1175. Primers and plasmids used in this study are listed in *Table 3* and *Table 4*.

## Co-expression of ATP8B1 with CDC50A in yeast membranes

Yeasts were transformed using the lithium-acetate method (*Gietz et al., 1995*). Yeast cultures, recombinant protein expression and membrane preparation were performed as described previously (*Azouaoui et al., 2014*; *Azouaoui et al., 2016*). Briefly, yeast growth took place in a glucose-containing rich growth medium supplemented with 2.7% ethanol at 28 °C for 36 h, whereas expression of the proteins of interest took place during an additional 18 h in the presence of 2% galactose, at 18 °C. Yeast cells were harvested by centrifugation, washed first with ice-cold ddH$_2$O, then with ice-cold TEKS buffer (50 mM Tris-HCl pH 7.5, 1 mM EDTA, 0.1 M KCl, 0.6 M sorbitol), and finally resupended in TES buffer (50 mM Tris-HCl pH 7.5, 1 mM EDTA, 0.6 M sorbitol) supplemented with protease inhibitors. The cells were subsequently broken with 0.5 mm glass beads using a 'Pulverisette 6' planetary mill (Fritsch). The crude extract was then spun down at 1000 *g* for 20 min at 4 °C, to remove cell debris and nuclei. The resulting supernatant was centrifuged at 20,000 *g* for 20 min at 4 °C, yielding S2 supernatant and P2 pellet. The S2 supernatant was further centrifuged at 125,000 *g* for 1 hr at 4 °C. The resulting P2 and P3 pellets were finally resuspended at about 30–50 mg ml$^{-1}$ of total protein in TES buffer. P2 and P3 membrane fractions were pooled and the ATP8B1 content was estimated, by immunoblotting, to be about 0.5% of total proteins.

## Purification of the ATP8B1-CDC50A complex

Membranes obtained after co-expression of ATP8B1 and CDC50A (P2 +P3) were diluted to 5 mg ml$^{-1}$ of total protein in ice-cold buffer A (50 mM MOPS-Tris at pH 7, 100 mM NaCl, 1 mM DTT, 20% (w/v) glycerol and 5 mM MgCl$_2$), supplemented with 1 mM PMSF and an EDTA-free protease inhibitor mixture. The suspension was stirred gently on a wheel for 5 min at 4 °C. Washed membranes were pelleted by centrifugation at 100,000 *g* for 1 hr at 4 °C. For cryo-EM sample preparation, this step was omitted and the membranes were directly incubated with DDM as follows. The pelleted membranes were resuspended at 5 mg ml$^{-1}$ of total protein in ice-cold buffer A supplemented with 1 mM PMSF and the EDTA-free protease inhibitor mixture. A mixture of DDM and CHS at final concentrations of 15 mg ml$^{-1}$ and 3 mg ml$^{-1}$, respectively, was added, resulting in a DDM/protein ratio of 3/1 (w/w). The suspension was then stirred gently on a wheel for 1 hr at 4 °C. Insoluble material was pelleted by centrifugation at 100,000 *g* for 1 hr at 4 °C. The supernatant, containing solubilized proteins, was applied onto a streptavidin-sepharose resin and incubated for 2 hr at 6 °C to allow binding of the BAD-tagged ATP8B1 to the resin.

For structural studies the DDM/CHS mixture was exchanged to LMNG/CHS. The resin was washed twice with three resin volumes of ice-cold buffer B (50 mM MOPS-Tris at pH 7, 100 mM KCl, 1 mM DTT, 20% (w/v) glycerol and 5 mM MgCl$_2$), supplemented with 0.2 mg ml$^{-1}$ LMNG and 0.02 mg ml$^{-1}$ CHS in the presence of 1 mM PMSF and an EDTA-free protease inhibitor cocktail. The resin was then washed thrice with three resin volumes of ice-cold buffer B supplemented with 0.1 mg ml$^{-1}$ LMNG and 0.01 mg ml$^{-1}$ CHS. Elution was performed by addition of 60 µg of purified TEV per ml of resin and overnight incubation at 6 °C. The eluted fraction was concentrated using a Vivaspin unit (100 kDa MWCO) prior to injection on a size-exclusion Superose 6 10/300 GL increase column equilibrated with buffer C (50 mM MOPS-Tris pH 7, 100 mM KCl, 1 mM DTT, 5 mM MgCl$_2$, 0.03 mg ml$^{-1}$ LMNG and 0.003 mg ml$^{-1}$ CHS). This step allowed separation of the TEV protease from the ATP8B1-CDC50A complex. The ATP8B1-CDC50A-containing fractions were pooled, concentrated using a Vivaspin unit (50 kDa MWCO) to concentrate the protein and the detergent micelles, and supplemented with LMNG and PI(3,4,5)P$_3$ to final concentrations of 0.35 mg ml$^{-1}$ and 0.05 mg ml$^{-1}$, respectively (PI(3,4,5)P$_3$/LMNG ratio of 0.15). The sample was then incubated for 1 hr at room temperature and overnight at 6 °C to allow lipid diffusion prior injection on a Superose 6 10/300 GL increase column equilibrated with buffer C, to remove the excess of detergent/lipid micelles.

For functional studies, the resin was washed four times with three resin volumes of ice-cold buffer B supplemented with 0.5 mg ml$^{-1}$ DDM and 0.1 mg ml$^{-1}$ CHS in the presence of 1 mM PMSF and an EDTA-free protease inhibitor cocktail. Elution was performed by addition of 60 µg of purified TEV per mL of resin by overnight incubation at 6 °C. For purifying the 3 C protease site-containing version of

ATP8B1, 240 µg of purified 3 C protease per ml of resin were added together with the TEV protease. Purified ATP8B1-CDC50A complex was snap-frozen and stored at –80 °C. ATP8B1 protein concentrations were calculated based on Coomassie-blue staining of SDS-PAGE gels using known amounts of purified Drs2.

## Grid preparation for Cryo-EM

The ATP8B1-CDC50A complex at a concentration of 0.8 mg ml$^{-1}$ was supplemented with 1 mM BeSO$_4$ and 5 mM KF to stabilize an E2-BeF$_x$ form mimicking the E2P conformation. The sample was incubated on ice for 1 hr and 3 µl were added to freshly glow-discharged (45 s at 15 mA) C-flat Holey Carbon grids, CF-1.2/1.3–4 C (Protochips), which were subsequently vitrified at 4 °C and 100% humidity for 4.5 s with a blotting force of –1 on a Vitrobot IV (Thermo Fisher Scientific) with standard Vitrobot filter paper (ø55/20 mm, Grade 595).

## Cryo-EM data collection

The Data were collected on a Titan Krios G3i (EMBION Danish National cryo-EM Facility – Aarhus node) with X-FEG operated at 300 kV and equipped with a Gatan K3 camera and a Bioquantum energy filter using a slit width of 20 eV and with 30° tilt. Movies were collected using aberration-free image shift data collection (AFIS) in EPU (Thermo Fisher Scientific) as 1.5 s exposures in super-resolution mode at a physical pixel size of 0.66 Å/pixel (magnification of ×130,000) with a total electron dose of 60 e$^-$/Å$^2$. A total of 3941 movies were collected.

## Cryo-EM data processing

Processing was performed in cryoSPARC v3 (*Punjani et al., 2017*). Patch Motion Correction and Patch CTF were performed before low-quality micrographs (e.g. micrographs with crystalline ice, high motion) were discarded. Particles were initially picked using a circular blob on ~1000 micrographs. These were aligned in 2D to produce references for template picking on all movies. Particles were extracted in a 416-pixel box and Fourier cropped to a 104-pixel box (2.64 Å/pixel). *Ab initio* references were produced using a subset of all particles. One protein-like reference and multiple junk references were used in multiple rounds of heterogeneous refinement. Selected particles were then re-extracted in a 416-pixel box (0.66 Å/pixel) before non-uniform (NU) refinement (*Punjani et al., 2020*). The particle stack was then CTF-refined using Local CTF refinement and motion-corrected using Local motion correction before final non-uniform (NU) refinement. Data processing flow-chart is available in *Figure 2—figure supplement 1*.

## Model building

The ATP8B1-CDC50A model was built using a homology model of ATP8B1 generated by I-TASSER (*Yang et al., 2015*) with Drs2 E2P$_{autoinhibited}$ (PDB: 6ROH) and from the CDC50A structure of the ATP8A1-CDC50A complex in E2P (PDB: 6K7L) as templates. The cryo-EM map was sharpened with a B factor of –84 Å$^2$ using the Autosharpen tool in PHENIX (*Terwilliger et al., 2018*).

The model was manually generated and relevant ligands added with COOT (*Emsley et al., 2010*) before real space refinement in PHENIX (*Afonine et al., 2018*) with secondary structure restraints. Model validation was performed using MolProbity (*Chen et al., 2010*) in PHENIX (*Adams et al., 2010*), and relevant metrics are listed in *Table 1*. Representative map densities with fitted models can be seen in *Figure 2—figure supplement 2*. Figures were prepared in ChimeraX (*Pettersen et al., 2021*).

## Endoglycosidase treatment

For CDC50A deglycosylation, the purified sample was treated with EndoH-MBP according to manufacturer instructions. Briefly, about 1.5 µg of purified ATP8B1-CDC50A complex was denatured for 3 min at 96 °C in the presence of 0.5% SDS and 40 mM DTT, in a final volume of 19.5 µl. The denatured proteins were then supplemented with 500 U of EndoH-MBP (EndoHf, NEB) and incubated for 45 min at 37 °C. Then 20 µl of urea-containing Laemmli denaturation buffer were added and the samples were incubated for 10 min at 30 °C prior loading on an 8% SDS-PAGE.

## Determination of subunit stoichiometry

About 6.5 µg of purified ATP8B1-CDC50A complex was denatured for 5 min at 96 °C, in the presence of 0.5% SDS and 40 mM DTT and in a final volume of 250 µl. The denatured proteins were then supplemented with 750 U of EndoH-MBP and incubated for 1 h at 37 °C. Samples were then precipitated by adding 1 volume of 1 M trichloroacetic acid (TCA). After 45 min on ice, samples were centrifuged at 20,000 g for 25 min at 4 °C. Supernatant was discarded and samples were centrifuged again at 20,000 g for 5 min at 4 °C to remove traces of TCA. Pellets were then resuspended in 60 µl urea-containing Laemmli buffer (50 mM Tris-HCl pH 6.8, 0.7 M β-mercaptoethanol, 2.5% w/v SDS, 0.5 mM EDTA, 4.5 M urea, 0.005% w/v bromophenol blue). Thirty µl of each sample (about 3.25 µg of purified complex) were loaded on a 4–15% gradient TGX stain-free gel. After 90 min electrophoretic separation at 150 V and 40 mA, the gel was soaked in 5% (w/v) TCA for 10 min and rinsed three times in ddH$_2$O. The gel was then exposed to UV (254 nm) for 5 min and images were collected after 20 s of exposure. The relative intensity of ATP8B1 and CDC50A was quantified from various amounts loaded onto gradient TGX stain-free gels using the ImageJ software.

## Phosphorylation of ATP8B1-CDC50A by [γ-$^{32}$P]ATP

To study phosphorylation of the ATP8B1-CDC50A complex, about 0.5 µg of purified complex were supplemented with [γ-$^{32}$P]ATP at a final concentration of 2 µM (5 mCi µmol$^{-1}$) and incubated at 0 °C in buffer B supplemented with 0.5 mg ml$^{-1}$ DDM and 0.1 mg ml$^{-1}$ CHS. Phosphorylation was stopped after 30 s by addition of 1 sample volume of 1 M TCA, 5 mM H$_3$PO$_4$. Samples were then left for 40 min on ice for aggregation and 2 volumes of 0.5 M TCA in 2.5 mM H$_3$PO$_4$ were subsequently added to help aggregation. Proteins were then centrifuged at 14,000 g for 25 min at 4 °C. The supernatant was removed, and the pellet was washed by addition of 0.5 M TCA in 0.5 mM H$_3$PO$_4$. Samples were centrifuged again at 14,000 g for 25 min at 4 °C. Supernatants were discarded, samples were centrifuged again at 14,000 g for 5 min at 4 °C to remove residual TCA. Pellets were then resuspended at 4 °C in 25 µl urea-containing Laemmli denaturation buffer. After resuspension, 15 µl of each sample (about 0.3 µg of purified complex) were loaded on acidic gels. The stacking gel contained 4% acrylamide, 65 mM Tris-H$_3$PO$_4$ pH 5.5, 0.1% SDS, 0.4% ammonium persulfate, and 0.2% TEMED, and the separating gel was a continuous 7% gel containing 65 mM Tris-H$_3$PO$_4$ pH 6.5, 0.1% SDS, 0.4% ammonium persulfate, and 0.05% TEMED. The gel tanks were immersed in a water/ice bath and the pre-cooled running buffer contained 0.1% SDS and 170 mM MOPS-Tris at pH 6.0. Dried gels were subsequently stained with Coomassie Blue before radioactivity was measured, using a PhosphorImager equipment (Amersham Typhoon RGB, GE Healthcare).

## ATPase activity of purified ATP8B1-CDC50A

For the ATP8B1-CDC50A complex, the rate of ATP hydrolysis was monitored continuously on an Agilent 8,453 diode-array spectrophotometer, using an enzyme-coupled assay. ATPase activity was measured at either 30 °C or 37 °C in buffer B supplemented with 1 mM ATP, 1 mM phosphoenolpyruvate, 0.4 mg ml$^{-1}$ pyruvate kinase, 0.1 mg ml$^{-1}$ lactate dehydrogenase, 250 µM NADH, 1 mM NaN$_3$, 1 mg ml$^{-1}$ DDM (2 mM), and residual CHS at 0.01 mg ml$^{-1}$. In these experiments, 50–200 µl of the purified ATP8B1-CDC50A complex (final concentrations of 1–5 µg ml$^{-1}$) was added to a total volume of 1.8 ml. For measurement of the half-maximum inhibitory concentration (IC$_{50}$), successive additions of the C-terminal peptide or its phosphorylated derivative (from a 1 mM stock solution) to purified ATP8B1-CDC50A incubated in 43 µg ml$^{-1}$ POPC ( ~ 57 µM), 25 µg ml$^{-1}$ PI(4,5)P$_2$ ( ~ 23 µM) and 0.5 mg ml$^{-1}$ DDM ( ~ 1 mM) in the assay cuvette were performed. Similarly, to determine the maximum rate of ATP hydrolysis ($V_{max}$) and the apparent affinity ($K_m$) for PPIns, successive additions of DDM and POPC to purified ATP8B1-CDC50A preincubated with 43 µg ml$^{-1}$ POPC, 25 µg ml$^{-1}$ PI(4,5)P$_2$ and 0.5 mg ml$^{-1}$ DDM were performed, in order to gradually decrease the PIP/DDM ratio (while the POPC/DDM ratio remained constant). Conversion from NADH oxidation rates expressed in mAU s$^{-1}$ to ATPase activities expressed in µmol min$^{-1}$ mg$^{-1}$ was based on the extinction coefficient of NADH at 340 nm ( ~ 6.2 mM$^{-1}$ cm$^{-1}$). For all experiments, photobleaching of NADH was reduced by inserting an MTO J310A filter that eliminates short wavelength UV excitation light. This setup reduced the spontaneous rate of NADH absorption changes down to ~0.01 mAU s$^{-1}$. ATPase activities measured for truncated ΔC1174 and ΔN42/C1174 come from two independent purification batches, with similar results, and referred to as 'purification #1' and 'purification #2' in the legend to figures.

## Quantification of n-dodecyl-β-ᴅ-maltoside bound to the transmembrane domain of purified Drs2-Cdc50

The yeast Drs2-Cdc50 flippase complex was purified by streptavidin-affinity chromatography, as previously described (*Azouaoui et al., 2017*). The complex was eluted in a buffer containing 50 mM MOPS-Tris pH 7, 100 mM KCl and 5 mM $MgCl_2$, supplemented with 0.5 mg ml⁻¹ DDM, and concentrated to about 1 mg ml⁻¹ on YM100 Centricon units (Millipore). Next, the eluted complex was supplemented with radioactive detergent ($^{14}$C-DDM, Commissariat à l'Énergie Atomique et aux Énergies Alternatives, Saclay) as a tracer, in order to evaluate the amount of DDM bound to the complex. A TSK3000 SW column (Tosoh Bioscience, Germany) was first equilibrated with 1 volume of 50 mM MOPS-Tris pH 7, 100 mM KCl, 5 mM $MgCl_2$ supplemented with 0.5 mg ml⁻¹ DDM, at room temperature. A second volume of mobile phase was applied, now supplemented with $^{14}$C-DDM. Both the purified complex and the mobile phase contained $^{14}$C-DDM to reach a specific activity of about $3.10^{-5}$ µCi per nmol of DDM. Fractions of 250 µl eluting between 5 ml and 10 ml were collected. Protein and $^{14}$C-DDM contents were determined by the bicinchoninic assay and liquid scintillation, respectively.

## HRV 3C protease purification

*Escherichia coli* (BL21) cells transformed with a $His_6$-$Arg_8$-GST-3C protease coding sequence cloned into pGEX-4T-2 plasmid were cultured in LB medium containing 100 µg L⁻¹ ampicillin and 30 µg L⁻¹ chloramphenicol. Protein expression was induced with 0.2 mM isopropyl-β-ᴅ-1-thiogalactopyranoside for 16 hr at 18 °C. Cells were harvested and lysed in lysis buffer C (50 mM $NaH_2PO_4$ pH 8, 500 mM NaCl, 30 mM imidazole, 10% glycerol (v/v) and 5 mM β-mercaptoethanol) by sonication. Cell debris were removed by centrifugation at 15,000 *g* for 30 min at 4 °C. The clarified lysate was loaded onto a HisTrap FF crude column (GE). To remove impurities, the column was washed with 6 column volumes of lysis buffer C followed by 15 column volumes of washing buffer D (50 mM $NaH_2PO_4$ pH 8, 150 mM NaCl, 30 mM imidazole, and 5 mM β-mercaptoethanol). The protein of interest was eluted with a gradient of elution buffer E (50 mM $NaH_2PO_4$ pH 8, 150 mM NaCl, 500 mM imidazole, and 5 mM β-mercaptoethanol). Fractions of interest were diluted two-fold and loaded onto a GST-Trap HP column. To remove impurities, the column was washed with 10 column volumes of GST-washing buffer F (8 mM $Na_2HPO_4$, 1.5 mM $KH_2PO_4$ pH 7.2, 140 mM NaCl, 2.7 mM KCl, 0.1 mM EDTA, 1 mM DTT). The protein of interest was eluted with a gradient of GST-washing buffer F supplemented with 40 mM of reduced glutathione. The fraction of interest was directly loaded onto a SP Sepharose Fast-Flow HiTrap column pre-equilibrated in buffer G (50 mM $NaH_2PO_4$ pH 8, 100 mM NaCl, 0.1 mM EDTA and 1 mM DTT). The column was washed with 5 column volumes of buffer D and the protein of interest was eluted with a gradient of buffer H (50 mM $NaH_2PO_4$ pH 8, 1.5 M NaCl, 0.1 mM EDTA and 1 mM DTT). Fractions containing the protein of interest were loaded on a HiLoad 16/600 Superdex 200 column pre-equilibrated in buffer I (50 mM MOPS-Tris pH 7, 100 mM KCl, 20% (w/v) glycerol and 1 mM DTT). Fractions containing the 3 C protease were pooled, concentrated to 3 mg ml⁻¹, aliquoted, snap-frozen and stored at –80 °C.

## TEV protease purification

*Escherichia coli* C43 (DE3) cells transformed with a MBP-TEV$_{site}$-His$_7$-TEV$_{S219V}$-Arg$_5$ protease coding sequence cloned into the pRK793 plasmid were cultured in LB medium containing 100 µg L⁻¹ ampicillin. Protein expression was induced with 0.5 mM isopropyl-β-ᴅ-1-thiogalactopyranoside for 16 h at 18 °C. Cells were harvested and lysed in lysis buffer J (50 mM Tris-HCl pH 7.5, 300 mM NaCl, 10% v/v glycerol) by sonication. Cell debris were removed by centrifugation at 10,000 *g* for 20 min at 4 °C. The clarified lysate was loaded onto a HisTrap FF crude column (GE). To remove impurities, the column was washed with 6 column volumes of lysis buffer J followed by 25 column volumes of washing buffer K (50 mM Tris-HCl pH 7.5, 300 mM NaCl, 10% v/v glycerol v/v, 25 mM imidazole). The protein of interest was eluted with a gradient of elution buffer L (50 mM Tris-HCl pH 7.5, 300 mM NaCl, 10% v/v Glycerol, 500 mM imidazole). Fractions of interest were diluted threefold in buffer M (50 mM $KH_2PO_4$ pH 8, 0.1 mM EDTA and 1 mM DTT) and loaded to a SP Sepharose Fast-Flow HiTrap column pre-equilibrated in buffer N (50 mM $KH_2PO_4$ pH 8, 100 mM NaCl, 0.1 mM EDTA and 1 mM DTT). The column was washed with 10 column volumes of buffer N. The protein of interest was eluted with a gradient of buffer O (50 mM $KH_2PO_4$ pH 8, 1.5 M NaCl, 0.1 mM EDTA and 1 mM DTT). Fractions containing the protein of interest were loaded on a HiLoad 16/600 Superdex 200 column

pre-equilibrated in buffer P (50 mM Tris-HCl pH 7.5, 200 mM NaCl). Elution fractions containing the TEV protease were pooled, supplemented with 30% glycerol (v/v), concentrated to 1 mg ml$^{-1}$, aliquoted, snap-frozen and stored at –80 °C.

## Statistical analysis, curve fitting and equations used in this tudy

Statistical analysis and curve fitting was carried out with the GraphPad Prism 9 software, and statistical significance was assigned to differences with a p value of < 0.05.

GraphPad Prism log (inhibitor) vs response-variable slope (four parameters) non-linear regression analysis was used to fit data displayed in *Figure 5B and C* and *Figure 4—figure supplement 2B*. This non-linear regression model is given by:

Y = Bottom + (Top-Bottom)/(1 + 10^((LogIC50 – X)*HillSlope)), where Y is the expected response, Top and Bottom are plateaus in the unit of the Y axis, IC50 is the concentration of peptide (or BeFx for *Figure 4—figure supplement 2B*) that gives a response halfway between Top and Bottom, and HillSlope is the slope at the steepest part of the curve, also known as the Hill slope.

GraphPad Prism Michaelis-Menten non-linear regression analysis was used to fit data displayed in *Figure 6C* and *Figure 4—figure supplement 2C*. This non-linear regression model is given by:

Y = Vmax*X/(Km +X), where Vmax is the maximum velocity in the same unit as Y and Km is the Michaelis-Menten constant, in the same units as X. Km is the substrate concentration needed to achieve a half-maximum enzyme velocity.

## Acknowledgements

We thank Thomas Boesen, Andreas Bøggild and Taner Drace for technical support during EM data collection at the EMBION Danish National cryo-EM facility of Aarhus University (5072-00025B, Danish Agency for Research and Higher Education), Jesper Lykkegaard Karlsen for scientific computing support, Joost Holthuis (University of Osnabruck, Germany) for kindly providing the ATP8B1 and CDC50A cDNAs, Rosa López-Marqués (University of Copenhagen, Denmark) for the gift of the *S. cerevisiae Δpep4* strain and Mads Eskesen Christensen and Natalya Fedosova for generously providing the pig kidney α1β1 isoform of Na$^+$/K$^+$-ATPase. We also wish to thank David Stokes for critical reading of the manuscript and Philippe Champeil, Alenka Čopič, Guillaume Drin, Rasmus Kock Flygaard, Francis Haraux, Anaïs Lamy, José Luis Vázquez-Ibar and Marc le Maire for discussion and advice.

## Additional information

### Funding

| Funder | Grant reference number | Author |
|---|---|---|
| EMBO | Short-term fellowship 7881 | Thibaud Dieudonné |
| French Infrastructure for Integrated Structural Biology | FRISBI ANR-10-INSB-05 | Christine Jaxel Cédric Montigny Guillaume Lenoir |
| French Ministry for Higher Education | PhD fellowship | Thibaud Dieudonné |
| European Commission | Marie Sklodowska-Curie individual fellowship | Thibaud Dieudonné |
| Agence Nationale de la Recherche | Young investigator grant ANR-14-CE09-0022 | Guillaume Lenoir |
| Lundbeckfonden | Professorship grant | Poul Nissen |
| Deutsche Forschungsgemeinschaft | GU 1133/11-1 | Thomas Günther Pomorski |
| Danish Agency for Science and Higher Education | 5072-00025B - Danish National Cryo-EM Research Infrastructure (EMBION) | Poul Nissen |

| Funder | Grant reference number | Author |
|---|---|---|

The funders had no role in study design, data collection and interpretation, or the decision to submit the work for publication.

## Author contributions

Thibaud Dieudonné, Formal analysis, Funding acquisition, Investigation, Methodology, Supervision, Validation, Visualization, Writing - original draft, Writing – review and editing; Sara Abad Herrera, Formal analysis, Investigation, Methodology, Writing – review and editing; Michelle Juknaviciute Laursen, Maylis Lejeune, Formal analysis, Investigation; Charlott Stock, Formal analysis, Visualization, Writing – review and editing; Kahina Slimani, Investigation; Christine Jaxel, Formal analysis, Investigation, Writing – review and editing; Joseph A Lyons, Formal analysis, Resources, Visualization, Writing – review and editing; Cédric Montigny, Formal analysis, Investigation, Supervision, Validation, Writing – review and editing; Thomas Günther Pomorski, Poul Nissen, Conceptualization, Formal analysis, Funding acquisition, Supervision, Validation, Writing – review and editing; Guillaume Lenoir, Conceptualization, Formal analysis, Funding acquisition, Investigation, Project administration, Supervision, Validation, Visualization, Writing - original draft, Writing – review and editing

## Author ORCIDs

Thibaud Dieudonné http://orcid.org/0000-0001-6988-4121
Charlott Stock http://orcid.org/0000-0001-5471-3696
Christine Jaxel http://orcid.org/0000-0002-1387-4458
Cédric Montigny http://orcid.org/0000-0003-0905-9861
Thomas Günther Pomorski http://orcid.org/0000-0002-4889-0829
Poul Nissen http://orcid.org/0000-0003-0948-6628
Guillaume Lenoir http://orcid.org/0000-0002-8759-5179

## Decision letter and Author response

Decision letter https://doi.org/10.7554/eLife.75272.sa1
Author response https://doi.org/10.7554/eLife.75272.sa2

# Additional files

## Supplementary files

• Transparent reporting form

## Data availability

Refined coordinates for the atomic model of the autoinhibited state of ATP8B1 have been deposited in PDB under the accession code 7PY4. The cryo-EM map of autoinhibited ATP8B1 has been deposited in EMDB under the accession code EMD-13711.

The following datasets were generated:

| Author(s) | Year | Dataset title | Dataset URL | Database and Identifier |
|---|---|---|---|---|
| Dieudonné T, Lenoir G, Nissen P | 2021 | Atomic model of ATP8B1-CDC50A in the E2P autoinhibited state | https://www.rcsb.org/structure/unreleased/7py4 | RCSB Protein Data Bank, 7PY4 |
| Dieudonné T, Lenoir G, Nissen P | 2021 | Cryo-EM map of ATP8B1-CDC50A in the E2P autoinhibited state | https://www.ebi.ac.uk/emdb/EMD-13711 | Electron Microscopy Data Bank, EMD-13711 |

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
