## [Editor Report]

This manuscript reports the first high-resolution structure of the P4 flippase ATP8B1, which is associated with intrahepatic cholestatic disorder in humans. Using biochemical studies guided by the structure, the authors demonstrate ATP8B1's autoinhibition mechanism, its regulation by lipids and phosphorylation, and a plausible mechanism of disease-associated mutation. These results are an important contribution to the expanding literature on membrane protein dynamics and function.

---

## [Decision Letter]

**Decision letter after peer review:**

Thank you for submitting your article "Autoinhibition and regulation by phosphoinositides of ATP8B1, a human lipid flippase associated with intrahepatic cholestatic disorders" for consideration by *eLife*. Your article has been reviewed by 3 peer reviewers, and the evaluation has been overseen by a Reviewing Editor and Richard Aldrich as the Senior Editor. The following individual involved in review of your submission has agreed to reveal their identity: Kazuhiro Abe (Reviewer #3).

Essential revisions:

1) Looking at the structure files, Glu20 in the N-term makes a hydrogen bond with Ser598 in the N-domain. Because the C-terminal tail is covered by this part of N-domain that seems to prevent C-terminal dissociation, this interaction between N-terminal tail and the N-domain may be important to describe its synergy effect on autoinhibition. The figure panel to show that interaction, together with the relative orientations of N-term, C-term and N-domain may be helpful for reader to follow its synergy effect. Please consider to make such a figure in the main text.

2) Please show the detailed molecular interactions between C-term and cytoplasmic domains. Although EM density and side chains are shown in Figure 3, the number of amino acids, and hydrophobic/hydrophilic interactions with surroundings are not readily understood from this figure. This is not necessarily as a ribbon-stick models. For example, 2D-diagram may also be useful to follow which residue(s) of C-term peptide binds to the A, P domains and the N domain, holding them together like glue.

3) Does the binding of C-terminal tail occur conformational dependent? The EM structure clearly visualizes the C-terminal tail bound in between the cytoplasmic domains arranged in the E2P state. Does this mean the autoinhibition always occur in the E2P state?

4) Is it possible to explain mechanistically how autoinhibition by the C-term is removed upon PIPs binding to the TM binding pocket? It is feasible that the negatively-charged phosphate group of PIPs could bind to the positively-charged pocket. But what kind of conformational change or other molecular event could be expected to remove the autoinhibition?

5) Sequence comparison of the C-terminal tail is extensively discussed, but less described for the N-terminal tail. Is it because N-terminal sequence is specific for ATP8B1?

6) Concerning the activation by in vitro truncation of the C-terminus: do the authors believe the C-terminus is proteolytic degraded during activation in vivo? If so, the activation would be irreversible? However, studies on the yeast homolog Drs2p seem to suggest the activation/inhibition is reversible.

7) Concerning the conclusion of phosphoinositides being an activator. Very high concentration phosphoinositides were used, which raises some doubt. Have the authors considered the possibility that phosphoinositides are a substrate rather than an activator, as suggested by a similar study posted in BioRxiv [https://doi.org/10.1101/2021.11.07.467649]. On the other hand, the authors' inability to observe the PI(3,4,5)P3 density does seem to implicate an activation role, as the compound may diffuse away after activation, in a manner similar to the PI4P activation of the yeast Drs2 flippase. Please discuss all the possibilities.

8) Related to point 7: Adding a section discussing on physiologically relevant big picture of this study will develop a good story. For example, a discussion on the comparison of physiological concentrations of phospholipid with the concentrations needed to stimulate ATPase activity of ATP8B1 will be useful. Kinetic parameters (Km, Vm) could inform this discussion, and sections on regulation by phosphorylation and phosphoinositides are surely useful. In addition, there are a number of experiments that could strengthen the conclusions including transfecting ATP8B1 and CDC50 in HEK cells and artificially increasing PIP2 levels in the cells, expected to release the auto inhibition by disengaging C-terminal and increase in the rate of ATP hydrolysis, will be useful. Measurements of intracellular ATP levels will confirm the regulation phenomenon and support its physiological relevance. Similarly, adding the disease perspective by modulating functional levels of ATP8B1 would be important to complete the story. While we recognize that the suggested experiments may be outside the scope of the current manuscript, we would like to see a discussion of the issues.

9) Figure 1 – gel in panel C shows the purification of ATP8B1-CDC50A according to the legend, however it does not indicate the corresponding band/s for CDC50A. Multiple bands of CDC50 were confirmed by immunoblotting by His probe in supplemental figure 1C. Please indicate CDC50 on the gel in panel C of Figure 1. Please mention in the legend if the amount of Lane Mb and Lane Estrep is same.

10) Figure 1 —figure supplement 1. As a suggestion, the purification procedure in panel B could have included eluting the complex with addition of DesBiotin, instead of eluting it with TEV cleavage. In panel C, the band for ATP8B1 is really diffused. Clear and distinct band as shown in panel D is better. The treatment of EndoH also resulted in consolidation of bands for ATP8B1 near 140kD. Please mention in the text. Panel D does not show EndoH cleavage of 1.5ug protein sample, which is shown in the original source gel file. Please explain it in the legend. Mean values are included for panel E in the source file, please include original data points for densitometry in the source file.

11) Results section does not describe interactions between ATP8B1 and CDC50A while describing the architecture of ATP8B1 (on page 7) and the structural role of CDC50A in ATP8B1 function.

12) Figure 3: please show how interactions (electrostatic vs hydrophobic) are mediated between N- and C- terminal regions and specific residues in A/P/N domain. Also describe these interactions in the text.

13) The gel for eluted purified samples in Figure 4A are smeary for mutant ATP8B1-CDC50 complexes and do not indicate CDC50. Assuming that these mutant complexes were further purified by gel filtration chromatography and then used for ATPase assays in Figure 4B, please show the gel for purified complexes after gel filtration. The purity of mutant ATP8B1-CDC50 complexes shown in Figure 4A may not be sufficient enough to perform ATPase assays in Figure 4B. It will be good to show a gel as shown in panel D of Figure1—figure supplement 1 for these mutant complexes. Please mention the yield of these mutants in comparison to wild type in the text.

14) Figure 4B: Please include controls with just lipids and no protein and control without lipids. Please explain baseline corrections in the legend. The legend includes that data is mean of 3 replicate experiments (technical). Please include the information regarding biological (N) repeats.

15) Figure 5: The order of panels can be changed. Panel C can be panel A. Panel A can be panel B. Panel D can be Panel C and Panel B can be Panel D. Please show interaction of peptide with other residues in N- and A- domain. Please include control with just peptide in panel A. Please provide a table for IC50 values for panel B. Legend for panel B mentions "33 to 47 data points". Source file has 33 and 35 data points for C-ter and phosphorylated C-ter peptide respectively. Please correct. The legend also mentions that "mean and sd of 3-4 replicates". Providing source data files is sincerely appreciated, however, source data files are missing a few data points for Figure 5A, 5D. Many of these data points in source data file have single points. And the transparency statement mentions "no data was excluded". So, the question is why these data points are missing in these files. Please explain any technical details in this regard. Source data file is included for Figure S5C, but the figure is not included.

16) In general, please detail out baseline corrections and normalizing data in the legend for Figure 4, S4, 5, 6. Also mention IC50 values with 95% confidence interval in the legend.

17) Figure 6: Source data files are missing a few data points for Figure 6C. Many of these data points in source data file have single points. Please explain.

18) Line 301: "ATP8B1-CDC50A required PC and PI(4,5)P2 for enzyme turnover". Is it essentially required? Result in Figure 4B can be simply interpreted as a stimulation of basal ATPase activity in the presence of PI(4,5)P2 and in the absence of N- and C-terminal regions. A control for basal ATPase activity of ATP8B1-CDC50A in the absence of lipid (mentioned above) would be important to add in Figure 4B as well as in Figure 6A.

19) For Figure 8, mutations can be classified and colored differently according to different diseases (PFIC1, BRIC1 or ICP1) or based on their location. D554N and H535L are proposed to affect ATP binding. Ideally, the authors would consider testing mutants in the ATPase assay. Similarly, Line 457 mentions about basic residues predicted to bind phospholipid. Mutagenesis experiments to confirm the location of the lipid binding site and to evaluate how changes in the lipid translocation pathway affect stimulation of ATPase activity by phospholipid could be of value. These suggested experiments would provide possible explanation for structural basis of these effects and would make the reading more interesting.

---

## [Author Response]

Essential revisions:1) Looking at the structure files, Glu20 in the N-term makes a hydrogen bond with Ser598 in the N-domain. Because the C-terminal tail is covered by this part of N-domain that seems to prevent C-terminal dissociation, this interaction between N-terminal tail and the N-domain may be important to describe its synergy effect on autoinhibition. The figure panel to show that interaction, together with the relative orientations of N-term, C-term and N-domain may be helpful for reader to follow its synergy effect. Please consider to make such a figure in the main text.

We thank the reviewers for this insightful remark. We have now included a new version of Figure 3 to highlight the interaction of the N- and C-termini with the cytosolic domains of ATP8B1. Panel B of new Figure 3 describes detailed interaction of the N-terminal tail with the cytosolic domains of ATP8B1 while panel C describes detailed interactions of the C-terminal tail with the cytosolic domains. Figure 3 from the previous version of the manuscript has been added as Figure 3 —figure supplement 1. The main text has been modified accordingly, to better describe the types of interactions involved. Furthermore, the discussion has been slightly remodeled to take into account the contacts that the N-terminal tail makes with the cytosolic domains in the context of ATP8B1 inhibition by its C-terminal tail.

2) Please show the detailed molecular interactions between C-term and cytoplasmic domains. Although EM density and side chains are shown in Figure 3, the number of amino acids, and hydrophobic/hydrophilic interactions with surroundings are not readily understood from this figure. This is not necessarily as a ribbon-stick models. For example, 2D-diagram may also be useful to follow which residue(s) of C-term peptide binds to the A, P domains and the N domain, holding them together like glue.

We agree with the reviewers that displaying in more details the molecular interactions between the C-terminal tail of ATP8B1 and the cytosolic A-, N-, and P-domains would improve this manuscript and greatly benefit to the reader. Hence, Figure 3 has been modified to highlight interactions of the C-terminal tail with the three cytosolic domains (Figure 3C). We chose to display molecular interactions as a ribbon-stick model because 2D-diagrams appeared too dense to be used in a comprehensible manner.

3) Does the binding of C-terminal tail occur conformational dependent? The EM structure clearly visualizes the C-terminal tail bound in between the cytoplasmic domains arranged in the E2P state. Does this mean the autoinhibition always occur in the E2P state?

This is a very interesting issue. Although we cannot precisely answer this question with our own structural data – we only obtained the structure of ATP8B1 in the E2P state – we may provide elements that argue in favor of autoinhibition occurring in a conformation-dependent manner. First, for all P4-ATPases which structure has been determined so far, the C-terminal tail has been found in its autoinhibitory position (i.e. in a cleft between the three cytosolic domains) only in the E2P state. In this regard, previous work from Hiraizumi and colleagues is very informative, as they obtained structures of the full-length flippase complex ATP8A1-CDC50A in a number of different conformations, capturing an almost complete catalytic cycle of the P4-ATPase (Hiraizumi et al., (2019) *Science*, 365:1149). In ATP8A1-CDC50A, which is also subject to regulation by its C-terminus, the inhibitory C-terminus is observed only in the BeFx-stabilized E2P form and is completely disordered in other conformations, suggesting that autoinhibition specifically occurs in the E2P state. Second, we show in Figure 1D that full-length ATP8B1 may be phosphorylated from [γ-^32^P]ATP, indicating that in the E1 state, the presence of the C-terminal tail does not prevent accessibility of the nucleotide-binding site. As such, we envision the C-terminal tail in equilibrium between a state bound to the ATP8B1 cytosolic domains and an unbound state, this equilibrium being poised toward the bound state in the E2P conformation. The possibility that autoinhibition occurs in a conformation-dependent manner is now alluded to in the Discussion section.

4) Is it possible to explain mechanistically how autoinhibition by the C-term is removed upon PIPs binding to the TM binding pocket? It is feasible that the negatively-charged phosphate group of PIPs could bind to the positively-charged pocket. But what kind of conformational change or other molecular event could be expected to remove the autoinhibition?

We thank the reviewers for pointing this out. We actually believe our data do not allow us to conclude that autoinhibition by the C-terminal tail is removed upon phosphoinositide binding to the TM binding pocket. As can be seen in Figure 4B, the activity of the intact full-length ATP8B1 (‘WT’) is not stimulated by addition of PI(4,5)P_2_ (compare blue and open bars). The C-terminus of ATP8B1 must be removed for PI(4,5)P_2_ to exert its stimulatory effect. This result rather suggests that phosphoinositides mediate their activatory effect through a distinct mechanism that does not involve the tails, e.g. by promoting conformational changes in the membrane domain that could for instance regulate access to the substrate-binding site. A non-exclusive possibility could be that phosphoinositides participate in autoinhibition relief, as proposed in the case of the yeast Drs2-Cdc50 flippase complex (Timcenko et al., (2019) *Nature*, 571:366). However, binding of PI(4)P to Drs2 is concurrent with the ordering of an amphipathic helix just after TM10 (the amphipathic helix is not formed in the absence of PI(4)P), resulting in a mechanical destabilization of the autoinhibitory domain interactions with the cytosolic domains. For ATP8B1, the corresponding amphipathic helix is already present in the absence of any phosphoinositide bound, raising the possibility that the mechanism involved for autoinhibition relief is different from that of Drs2. Along those lines, the role of PPIns on the activation of ATP8B1 with C-terminal or double N- and C-terminal truncation could be interpreted as supporting a model where regulatory PPIns bind to the N-terminal tail of ATP8B1. We note with interest that the N-terminal tail contains a cluster of positively charged residues between P42 and D70 (including R46, R49, R55, R59 and K60) perhaps mediating activation of ATP8B1 by phosphoinositides. To make it clearer, this aspect of the mechanism by which phosphoinositides regulate ATP8B1-CDC50A activity is now mentioned in the Discussion section.

5) Sequence comparison of the C-terminal tail is extensively discussed, but less described for the N-terminal tail. Is it because N-terminal sequence is specific for ATP8B1?

The N-terminus sequence of ATP8B1 is indeed poorly conserved, even among other class 1 P4-ATPases (ATP8B2, ATP8B3, ATP8B4, ATP8A1, and ATP8A2) or other flippases known to be regulated by their termini (ATP8A1, ATP8A2, Drs2), and is therefore specific for ATP8B1. In contrast, the C-terminus is partially conserved as discussed in the article. Interestingly, Drs2 is also known to be regulated by its N-terminus (Azouaoui et al., (2017) *J Biol Chem*, 292:7954) but no unassigned density could be observed in the published cryo-EM map of full-length and autoinhibited Drs2-Cdc50 complex (EMDB: EMD-20468). Therefore, it remains unclear whether this interaction is specific to this flippase or if other flippase N-termini could interact similarly with the cytosolic domains of P4-ATPases.

6) Concerning the activation by in vitro truncation of the C-terminus: do the authors believe the C-terminus is proteolytic degraded during activation in vivo? If so, the activation would be irreversible? However, studies on the yeast homolog Drs2p seem to suggest the activation/inhibition is reversible.

As pointed out by the reviewers, proteolytic degradation would result in an irreversible activation of ATP8B1. One example of such regulatory proteolytic degradation has been previously described for a P4-ATPase, ATP11C, but in contrast to the regulatory mechanism discussed here for ATP8B1, it was inactivating proteolysis that occurred in the context of apoptosis (Segawa et al., (2014) *Science*, 344:1164). For ATP8B1, we do not suggest such a mechanism. Instead, activation could be mediated by the recruitment of protein partners to the N- and/or C-terminal tail which would sequester the tails to prevent autoinhibition, as previously suggested for the yeast Drs2-Cdc50 flippase (Timcenko et al., (2019) *Nature*, 571:366). In addition, the inhibitory properties of a peptide derived from the C-terminus of ATP8B1 suggest that phosphorylation of residue S1223 may provide additional regulation.

7) Concerning the conclusion of phosphoinositides being an activator. Very high concentration phosphoinositides were used, which raises some doubt. Have the authors considered the possibility that phosphoinositides are a substrate rather than an activator, as suggested by a similar study posted in BioRxiv [https://doi.org/10.1101/2021.11.07.467649]. On the other hand, the authors' inability to observe the PI(3,4,5)P3 density does seem to implicate an activation role, as the compound may diffuse away after activation, in a manner similar to the PI4P activation of the yeast Drs2 flippase. Please discuss all the possibilities.

Thanks for pointing this out. We checked the manuscript posted on BioRxiv but could not find any reference to the role of phosphoinositides in the study conducted by Cheng and colleagues. As to the concerns of reviewers regarding the concentration of phosphoinositides required to elicit ATP8B1 activity, it should be noted that in Figure 6C, the concentration of lipid is in fact expressed as the molar ratio of phosphoinositide over detergent (here, DDM). Indeed, the most relevant parameter to describe the effect of a particular lipid concentration in the presence of a given concentration of detergent is the lipid/detergent ratio, which reflects the local lipid concentration within the detergent micellar phase, rather than the lipid concentration in the aqueous volume. In other words, the concentration of detergent used in the ATPase assay is a parameter of critical importance, as it will largely influence the concentration of lipid available (Montigny et al., (2017) *PLoS One*, 12:e0170481). Along those lines, we previously showed, for the yeast Drs2-Cdc50 complex, that apparent affinities measured at low DDM concentrations (0.75-2 mg/ml) and those obtained at significantly higher DDM concentrations (3-5 mg/ml) are essentially similar if lipid concentrations are expressed as lipid/DDM ratios (Figure 5B, Azouaoui et al., (2017) *J Biol Chem*, 292:7954).

With this in mind, the *K_m_* value for activation of ATP8B1 by PI(3,4,5)P_3_ (PIP3) is about 1.4x10^-3^ mol PIP3/mol DDM. Based on our own estimation of the number of DDM molecules surrounding the transmembrane domain of Drs2-Cdc50 using size-exclusion chromatography in the presence of ^14^C-labeled DDM, we estimate that the detergent belt around the transmembrane region of ATP8B1-CDC50A is composed of ~ 270 molecules of DDM. Taking into account the additional presence of two transmembrane helices contributed by Cdc50, this is in the same range as the amount of DDM bound to purified SERCA1a, a P-type ATPase from the P2 subfamily, as determined by MALDI-TOF mass spectrometry (Chaptal et al., (2017) *Sci Rep*, 7:41751).

A *K_m_* value of 1.4x10^-3^ mol PIP3/mol DDM corresponds to ~0.38 mol of PIP3 per 270 mol of DDM (or 0.14 mol%) in the immediate environment of ATP8B1-CDC50A, emphasizing the strong affinity of ATP8B1 for PIP3. The apparent affinity of ATP8B1 for PIP3 is actually stronger than that of Drs2 for PI4P (Author response image 1). Given that PI4P has clearly been identified as a regulator of Drs2, and not as a transport substrate (Azouaoui et al (2017) *J Biol Chem*, 292:7954; Timcenko et al (2021) *J Mol Biol*, 433:167062), we think it is unlikely that phosphoinositides are transport substrates for ATP8B1. Comparison of physiological PIP3 concentrations with PIP3 concentrations required to activate ATP8B1-CDC50A in detergent micelles is now included in the discussion section of the revised manuscript.

**Author response image 1. sa2fig1:** Comparison of ATP8B1-CDC50A and Drs2-Cdc50 affinities for phosphoinositides. The results displayed for ATP8B1-CDC50A correspond to those shown in Figure 6C while the results displayed for Drs2 are adapted from Figure 5B published in Azouaoui et al., (2017) *J Biol Chem* 292:7954.

8) Related to point 7: Adding a section discussing on physiologically relevant big picture of this study will develop a good story. For example, a discussion on the comparison of physiological concentrations of phospholipid with the concentrations needed to stimulate ATPase activity of ATP8B1 will be useful. Kinetic parameters (Km, Vm) could inform this discussion, and sections on regulation by phosphorylation and phosphoinositides are surely useful.

We addressed this comment of the reviewers in our answer to comment #7 above.

In addition, there are a number of experiments that could strengthen the conclusions including transfecting ATP8B1 and CDC50 in HEK cells and artificially increasing PIP2 levels in the cells, expected to release the auto inhibition by disengaging C-terminal and increase in the rate of ATP hydrolysis, will be useful. Measurements of intracellular ATP levels will confirm the regulation phenomenon and support its physiological relevance. Similarly, adding the disease perspective by modulating functional levels of ATP8B1 would be important to complete the story. While we recognize that the suggested experiments may be outside the scope of the current manuscript, we would like to see a discussion of the issues.

We thank the reviewers for pointing out this very interesting set of experiments. As previously mentioned in response to comment #4, PPIns are actually not sufficient to trigger ATPase activity in vitro: the tails also need to be removed. As we don’t know yet the conditions for the activation of ATP8B1 in vivo, e.g. how phosphorylation of S1223 in ATP8B1 is regulated, there is little chance that only raising the levels of PPIns will allow disengagement of the C-terminus and an increase of the ATPase activity. Besides, the suggested experiment would only work provided ATP8B1 uses a substantial and significant amount of ATP among all other ATP-consuming systems in the cell, which is likely not the case not least because other P-type ATPases (from the P2 subfamily for instance) are more abundant and have a higher turnover rate than ATP8B1, thereby limiting the chances of success. We mentioned in the discussion that modulating PPIn levels would certainly be of great interest, but subject for future studies. Therefore, although the suggested experiments are certainly of great interest to investigate in cellulo the role of PPIns in ATP8B1 activation, we feel this is outside the scope of the current manuscript.

9) Figure 1 – gel in panel C shows the purification of ATP8B1-CDC50A according to the legend, however it does not indicate the corresponding band/s for CDC50A. Multiple bands of CDC50 were confirmed by immunoblotting by His probe in supplemental figure 1C. Please indicate CDC50 on the gel in panel C of Figure 1. Please mention in the legend if the amount of Lane Mb and Lane Estrep is same.

Thanks for pointing this out. CDC50A is hard to detect by Coomassie blue staining, probably because of its heterogenous glycosylation, as for instance previously observed for ATP11C (Segawa et al., (2018) *J Biol Chem*, 293:2172).

The position of CDC50A has now been included in panel C of Figure 1.

The amount of protein loaded in lanes ‘Mb’ and ‘E_Strep_’ of Figure 1C is different. The amount of protein loaded in each lane is now indicated in the legend to Figure 1C.

10) Figure 1 —figure supplement 1. As a suggestion, the purification procedure in panel B could have included eluting the complex with addition of DesBiotin, instead of eluting it with TEV cleavage. In panel C, the band for ATP8B1 is really diffused. Clear and distinct band as shown in panel D is better. The treatment of EndoH also resulted in consolidation of bands for ATP8B1 near 140kD. Please mention in the text. Panel D does not show EndoH cleavage of 1.5ug protein sample, which is shown in the original source gel file. Please explain it in the legend. Mean values are included for panel E in the source file, please include original data points for densitometry in the source file.

The affinity of the biotinylated acceptor domain for streptavidin beads is too strong for desthiobiotin to sufficiently weaken the interaction and allow release of the target protein. Elution with desthiobiotin, commonly used for Strep-tagged proteins binding to streptactin resin, did not prove successful when we optimized the purification protocol for other targets than ATP8B1. Additionally, in the presence of desthiobiotin, we would also elute endogenously biotinylated yeast proteins, resulting in a less pure sample. Finally, adding a TEV cleavage site allows to get rid of the 10 kDa biotin acceptor domain. For those reasons, we chose to insert a TEV-cleavable biotin acceptor domain.

In Figure 1 —figure supplement 1C, the band for ATP8B1 is diffuse upon treatment with EndoH (last lane). This is because as part of EndoH treatment, the purified ATP8B1-CDC50A complex is boiled before loading onto SDS-PAGE, which leads to aggregation of ATP8B1. This technical point is now mentioned in the legend to Figure 1 —figure supplement 1C. Additionally, ATP8B1 may appear more focused in panel D because in that case, precast 4-15% gels were used instead of our regular 8% acrylamide gels. Residual interaction of ATP8B1 with DDM during SDS-PAGE could also add to the diffuse electrophoretic mobility of ATP8B1. Such a behavior for DDM-purified proteins has already been observed, and this goes in line with the fact that ATP8B1 runs as a more focused band in panel D, where the proteins are precipitated with trichloroacetic acid (and hence the detergent removed) before loading onto gradient SDS-PAGE.

As to consolidation of ATP8B1 bands upon EndoH treatment, we believe the reviewers have been misled by the fact that boiling of ATP8B1 sample leads to some aggregation.

We now included lanes corresponding to EndoH cleavage of 1.5 µg protein, for consistency. Furthermore, regarding panel E of Figure 1 —figure supplement 1, densitometry values for original data points have now been included.

11) Results section does not describe interactions between ATP8B1 and CDC50A while describing the architecture of ATP8B1 (on page 7) and the structural role of CDC50A in ATP8B1 function.

We agree with the reviewers that a better description of ATP8B1-CDC50A interactions is required. The text of the revised manuscript was amended accordingly. Moreover, the structure of CDC50A is highly similar to what has already been described for other human flippase complexes in which it is present. To highlight this point, we included a structural alignment of CDC50A from the ATP8B1-CDC50A (this study), ATP8A1-CDC50A (PDB: 6K7L, Hiraizumi et al., 2019), ATP11C-CDC50A (PDB: 7BSU, Nakanishi et al., 2020) flippase complexes (Figure 2 —figure supplement 3). The contribution of the CDC50 subunit in P4-ATPase function, apart from its role in subcellular trafficking of the mature complex, is still elusive. Unfortunately, our data do not bring more information on this aspect than previous studies and we therefore preferred not to speculate on this matter.

12) Figure 3: please show how interactions (electrostatic vs hydrophobic) are mediated between N- and C- terminal regions and specific residues in A/P/N domain. Also describe these interactions in the text.

We addressed this request in our answer to comments #1 and #2.

13) The gel for eluted purified samples in Figure 4A are smeary for mutant ATP8B1-CDC50 complexes and do not indicate CDC50. Assuming that these mutant complexes were further purified by gel filtration chromatography and then used for ATPase assays in Figure 4B, please show the gel for purified complexes after gel filtration. The purity of mutant ATP8B1-CDC50 complexes shown in Figure 4A may not be sufficient enough to perform ATPase assays in Figure 4B. It will be good to show a gel as shown in panel D of Figure1—figure supplement 1 for these mutant complexes. Please mention the yield of these mutants in comparison to wild type in the text.

The gel for eluted purified samples in Figure 4A looks smeary for different reasons. First, proteolysis with 3C protease, meant to release the N-terminal tail (ΔN42) or the C-terminal tail (ΔC1174), or both (ΔN42/C1174), is not 100% complete. Therefore, the residual presence of full-length ATP8B1 for the mutants, in close proximity of truncated forms for ΔN42 and ΔC1174, makes the ATP8B1 band diffuse. Second, despite four consecutive purification steps (IMAC, glutathione sepharose, cation exchange and size-exclusion chromatography), the purified 3C protease is not pure to homogeneity. As we need substantial amounts for truncation of 3C constructs, addition of 3C protease in fact contaminates our purified ATP8B1-CDC50A sample. Third, it seems that destaining of the Coomassie blue stained gel is not homogenous. As requested, we included a legend to CDC50A in Figure 4A.

Note that the mutants, as well as the WT, were not purified by gel filtration before ATPase measurements. Indeed, the purification yield of ATP8B1-CDC50A is too scarce to envision systematic purification by gel filtration of streptavidin-purified samples. We will make this clearer in the text by providing the information in the legend to Figure 4A.

The reviewers are concerned by the fact that the lack of purity of the samples could prevent us to detect activity for the WT and ΔN42, for instance in the presence of PC and PI(4,5)P_2_. Our rationale is the following. While we cannot exclude that the activity of the WT and ΔN42 species is indeed overlooked because it is not significantly higher than the background noise, our point here is to highlight the importance of the N-terminal and C-terminal tails in autoinhibition. This remains true, even if we miss a minor activity of the WT and ΔN42. Moreover, rather than using end-point assay, we monitor the variations in NADH absorption (and therefore ATP consumption) continuously, improving the accuracy of the measurement of the ATP hydrolysis rate given by the slope of the various traces. Given the fact that the lack of purity of samples displayed in Figure 4B is mainly due to contamination by proteins present in the purified 3C sample, we may also argue that activity measurements of the intact WT ATP8B1-CDC50A complex that has not been treated with 3C do not reveal any stimulation by PC and PI(4,5)P_2_. In this case, the ATPase activity of ATP8B1-CDC50A is revealed thanks to limited proteolysis with trypsin in the ATPase assay cuvette (Author response image 2), in order to remove the autoinhibitory tails, and following a procedure devised for the yeast Drs2-Cdc50 complex (Azouaoui et al., (2017) *J Biol Chem*, 292:7954). This strongly suggests that if no activity can be revealed for WT and ΔN42 in Figure 4B, this is not because of their lack of purity.

**Author response image 2. sa2fig2:** ATPase activity measurements of streptavidin-purified WT ATP8B1-CDC50A. (A) ATPase activity of the purified ATP8B1-CDC50A complex determined in DDM/CHS at 30°C, using an enzyme-coupled assay, where the kinetics of NADH oxidation is monitored continuously. The various additions in the assay cuvette are indicated with arrows. Wild-type (WT) was added at ~ 2 µg ml^-1^ to continuously stirred cuvettes in an assay medium containing 1 mM MgATP, 0.5 mg ml^-1^ DDM, and 0.01 mg ml^-1^ CHS in buffer B. PC and PI(4,5)P_2_ were added at 0.1 mg ml^-1^ and 0.025 mg ml^-1^, respectively, resulting in a DDM final concentration of 1.25 mg ml^-1^. Trypsin and BeFx were added at 0.07 mg ml^-1^ and 1 mM, respectively. The rate of ATP hydrolysis corresponds to the slope measured after each addition. Activity is revealed upon addition of trypsin. (B) Specific ATPase activity of WT ATP8B1-CDC50A measured from traces such as that displayed in *(A)*. The dotted line represents the background NADH oxidation level, as measured before addition of ATP8B1-CDC50A in the assay cuvette. Data in (*B*) are a mean ± s.d. of 6 to 12 replicate experiments. PC: phosphatidylcholine.

Panel D of Figure 1 —figure supplement 1 displays a TGX gel specifically designed to reveal tryptophan fluorescence. Because those gels are 4-15% precast gels and detergent is stripped from samples upon TCA treatment, samples appear more focused. This method may not be sensitive enough for detecting the CDC50A band for the various mutants, for which the purification yield is lower. However, data displayed in Figure 4 —figure supplement 2 suggest that interaction of CDC50A with ATP8B1 is not compromised in truncation mutants.

The yield of the various mutants is now indicated in the text.

14) Figure 4B: Please include controls with just lipids and no protein and control without lipids. Please explain baseline corrections in the legend. The legend includes that data is mean of 3 replicate experiments (technical). Please include the information regarding biological (N) repeats.

As suggested by the reviewers, we added to Figure 4B ATPase activity measurements performed when the protein is present, but in the absence of lipids (light grey bars in Figure 4B). In the data displayed in Figure 4B, there is no baseline correction. On the contrary, to avoid missing any useful information associated with these experiments, we chose to plot data that have not been treated beforehand. The background NADH oxidation, measured before adding the purified ATP8B1-CDC50A complex and the various lipids, is symbolized by the presence of the dotted line in Figure 4B (see the legend to Figure 4B). When monitoring continuously at 340 nm the rate of ATP hydrolysis, via its coupling to NADH oxidation, we observe for all experiments photobleaching of NADH, resulting in the ‘background NADH oxidation’ mentioned above. We reduce this photobleaching of NADH by inserting an MTO J310A filter that eliminates the short wavelength UV-exciting light of our diode-array spectrophotometer lamp, as excitation of the 260-nm absorption band of NADH is much more deleterious for light stability of NADH than excitation of its 340-nm band. Furthermore, we are not completely sure why the reviewers are asking for a control with just lipids and no protein. We assume it is because the reviewers want to see the contribution of the various lipids added to the assay cuvette on the NADH oxidation rate. If so, Figure 4B already addresses this point as the addition of lipids to the WT complex, for instance, does not have any significant effect on the NADH oxidation rate. Incidentally, Figure 4B also shows that adding solely the purified complexes to the assay cuvette does not either promote NADH oxidation further than that corresponding to photobleaching. Therefore, this methodology allows to unambiguously conclude that the rate of ATP hydrolysis measured for the WT or ΔN42 in the absence or presence of lipids does not correspond to basal activity. In any case, we apologize for any confusion or misunderstanding of data displayed in this figure. To improve this, we suggest to add as Figure 4 —figure supplement 3 the raw absorption traces highlighting the various additions to the WT and catalytic D454N mutant of ATP8B1. The data plotted in Figure 4B are from one experiment, with technical replicates. However, we obtained similar results from two independent batches of purified ΔN42/C1174 and ΔC1174. Results from Figure 4B and Figure 5 reflect two independent (biological) replicates of purified ΔN42/C1174 and ΔC1174. We chose to plot data in Figure 4B from a single experiment because it allowed us to display values for specific activity while should we have combined results from two independent experiments, we would have needed to plot as % of max because of the non-negligible error associated with ATP8B1 quantification from SDS-PAGE. To mention clearly the reproducibility of the above-mentioned experiments, we stated in the Methods section that ΔN42/C1174 and ΔC1174 come from two independent batches. Furthermore, those two batches are referred to as ‘purification 1’ in the legend to Figure 4 and ‘purification 2’ in the legend to Figure 5.

15) Figure 5: The order of panels can be changed. Panel C can be panel A. Panel A can be panel B. Panel D can be Panel C and Panel B can be Panel D. Please show interaction of peptide with other residues in N- and A- domain. Please include control with just peptide in panel A. Please provide a table for IC50 values for panel B. Legend for panel B mentions "33 to 47 data points". Source file has 33 and 35 data points for C-ter and phosphorylated C-ter peptide respectively. Please correct. The legend also mentions that "mean and sd of 3-4 replicates". Providing source data files is sincerely appreciated, however, source data files are missing a few data points for Figure 5A, 5D. Many of these data points in source data file have single points. And the transparency statement mentions "no data was excluded". So, the question is why these data points are missing in these files. Please explain any technical details in this regard. Source data file is included for Figure S5C, but the figure is not included.

We thank the reviewers for this suggestion. We reorganized Figure 5 accordingly. A table reporting the IC50 values is also provided as Table 2. Moreover, an updated version of the structural details around S1223 showing the interactions of the C-terminal tail with residues of the A- and N-domain is now included in the manuscript.

By asking a control with just peptide in panel A, we assume the reviewers wonder whether high concentration of inhibitory C-terminal peptide would not interfere with the activity of the enzymes of the coupled assay and potentially compromise NADH oxidation or/and regeneration of ATP. To address this issue, we monitored absorbance at 340 nm of a cuvette containing the ATPase activity medium, in the absence or in the presence of the C-terminal peptide at 70 µM, a concentration that inhibits ATP8B1-CDC50A. Repeated additions of ADP at a final concentration of 10 µM led to the expected oxidation of 10 µM NADH, resulting in a fast OD change of ~0.06 AU at 340 nm. We conclude that the ATP8B1 C-terminal peptide has no adverse effect on the ability of the enzyme-coupled assay to regenerate ATP. This control experiment has been added to the manuscript as Figure 5 —figure supplement 1.

Regarding the number of data points used to plot IC_50_ values in panel B, 33 and 35 data points have indeed been used for C-ter and phosphorylated C-ter peptides against ΔN42/C1174 but 34 to 47 data points have been used for C-ter and phosphorylated C-ter peptides against ΔC1174, explaining why we mention “33 to 47 data points” in the legend to Figure 5B.

The number of replicates for each C-ter peptide concentration is not identical because the corresponding graphs have been plotted from three independent experiments, and the concentrations of peptide used for these three experiments were not exactly the same. On another note, for the fit to be as accurate as possible, we think the number of different peptide concentrations is important, as much as replicating exactly the same points from experiment to experiment. However, only three data points have single points and the rest of the points have been replicated 3 to 5 times. Therefore, we confirm that no data were excluded, as indicated in the transparency statement.

We apologize but we could not find any source data for Figure S5C. Therefore, we did not grasp what the reviewers are referring to.

16) In general, please detail out baseline corrections and normalizing data in the legend for Figure 4, S4, 5, 6. Also mention IC50 values with 95% confidence interval in the legend.

As mentioned in our response to comment #14, neither baseline correction nor normalization has been applied to data displayed in Figure 4. For Figure 4 —figure supplement 4A and 4B, the rate of ATP hydrolysis was corrected for NADH photobleaching occurring before the addition of the purified complex and ATP. We apologize for omitting this important precision in our first submission and this has now been corrected for. Additionally, for Figure 4 —figure supplement 4A, the activity in the presence of BeFx was normalized to the activity in the absence of BeFx, which was taken as 100%. For panels B and C of Figure 5, as the rate of ATP hydrolysis was corrected for the activity in the presence of BeFx, hence plotted is the BeFx-sensitive ATPase activity, and the activity in the absence of C-terminal peptide was taken as 100%. For panel E of Figure 5, the rate of ATP hydrolysis was corrected for photobleaching and the activity the activity in the absence of peptide was taken as 100% for each species. This is now clearly mentioned in the legend to Figure 5.

For panels A and B of Figure 6, the rate of ATP hydrolysis was corrected for NADH photobleaching occurring before the addition of the purified complex. For panel C of Figure 6, the BeFx-sensitive ATPase activity was plotted. No normalization has been applied to data displayed in Figure 6C. Finally, we included IC_50_ values with 95% confidence intervals in the legend to Figure 5B, 5C, and Figure 4 —figure supplement 4A.

17) Figure 6: Source data files are missing a few data points for Figure 6C. Many of these data points in source data file have single points. Please explain.

The data shown in panel C of figure 6 are the results of 3 to 4 independent experiments for which the concentrations of phosphoinositides were not necessarily identical, thereby explaining why there are for some concentrations only single points. But no data were excluded from these measurements.

18) Line 301: "ATP8B1-CDC50A required PC and PI(4,5)P2 for enzyme turnover". Is it essentially required? Result in Figure 4B can be simply interpreted as a stimulation of basal ATPase activity in the presence of PI(4,5)P2 and in the absence of N- and C-terminal regions. A control for basal ATPase activity of ATP8B1-CDC50A in the absence of lipid (mentioned above) would be important to add in Figure 4B as well as in Figure 6A.

In Figure 4B, the dotted line corresponds to background NADH oxidation, i.e. NADH bleaching, in the absence of any added protein or lipid. As can be seen from Figure 4B, the mere addition of the purified ATP8B1-CDC50A complex (condition ‘no lipid’) does not stimulate ATP hydrolysis, nor does the addition of PC alone. Therefore, now that activity in the absence of lipid is plotted, the results in Figure 4B cannot be interpreted as a stimulation of basal ATPase activity. As to Figure 6A, the dotted line indicates the background activity measured in the absence of added lipid.

19) For Figure 8, mutations can be classified and colored differently according to different diseases (PFIC1, BRIC1 or ICP1) or based on their location. D554N and H535L are proposed to affect ATP binding. Ideally, the authors would consider testing mutants in the ATPase assay. Similarly, Line 457 mentions about basic residues predicted to bind phospholipid. Mutagenesis experiments to confirm the location of the lipid binding site and to evaluate how changes in the lipid translocation pathway affect stimulation of ATPase activity by phospholipid could be of value. These suggested experiments would provide possible explanation for structural basis of these effects and would make the reading more interesting.

We fully agree that Figure 8 would be more attractive if mutations were classified and colored according to the nature of the disease and we did so in the new version of Figure 8.

The suggested mutagenesis experiments are certainly of very high interest but in practice, it’s not going to be possible to do that in a reasonable amount of time. The yield of purification of ATP8B1, although sufficient to conduct cryo-EM experiments, remains limiting for biochemical experiments. Purifying mutants requires several liters of culture for each mutant and we are actually working at the moment on improving yield and purity of ATP8B1 for the purpose of characterizing mutations found in PFIC1 and BRIC1 patients. This is currently the project of a PhD student in our lab. Unfortunately, the project is not mature enough to envision this can be done sufficiently quickly. Finally, the primary purpose of the manuscript we submit to *eLife* concerns the regulatory mechanism of ATP8B1-CDC50A, including autoinhibition by its terminal tails and activation by phosphoinositides, and we kindly ask for structure-function studies of the mutants to be kept for a follow-up article.